# Feedback inhibition underlies new computational functions of cerebellar interneurons

Hunter E Halverson[1†], Jinsook Kim[2,3†], Andrei Khilkevich[1], Michael D Mauk[1,4], George J Augustine[2,3]*

[1]Center for Learning and Memory, The University of Texas, Austin, United States; [2]Program in Neuroscience & Mental Health, Lee Kong Chian School of Medicine, Nanyang Technological University, Singapore, Singapore; [3]Institute of Molecular and Cell Biology, Singapore, Singapore; [4]Department of Neuroscience, The University of Texas, Austin, United States

**Abstract** The function of a feedback inhibitory circuit between cerebellar Purkinje cells and molecular layer interneurons (MLIs) was defined by combining optogenetics, neuronal activity recordings both in cerebellar slices and in vivo, and computational modeling. Purkinje cells inhibit a subset of MLIs in the inner third of the molecular layer. This inhibition is non-reciprocal, short-range (less than 200 μm) and is based on convergence of one to two Purkinje cells onto MLIs. During learning-related eyelid movements in vivo, the activity of a subset of MLIs progressively increases as Purkinje cell activity decreases, with Purkinje cells usually leading the MLIs. Computer simulations indicate that these relationships are best explained by the feedback circuit from Purkinje cells to MLIs and that this feedback circuit plays a central role in making cerebellar learning efficient.

*For correspondence: george.augustine@ntu.edu.sg

†These authors contributed equally to this work

Competing interest: The authors declare that no competing interests exist.

## Editor's evaluation

This is an important paper that describes and models an inhibitory pathway that mediates delay conditioning using cerebellar mechanisms in mice and rabbits. The manuscript provides convincing evidence for the proposed mechanisms, further supported by models.

## Introduction

Local inhibitory interneurons are ubiquitous throughout the brain and play diverse and fundamental roles in neuronal information processing (*Eccles et al., 1967*). Within the cerebellar cortex, Golgi, candelabrum, basket and stellate cells have been identified as inhibitory interneurons (*Hull and Regehr, 2022*). While Golgi cells are found in the granule cell layer and candelabrum cells are in the Purkinje cell (PC) layer, basket and stellate cells reside within the molecular layer of the cerebellar cortex. These molecular layer interneurons (MLIs) form inhibitory synapses with each other, as well as with PCs, the only output cells of the cerebellar cortex. Whereas specific computational roles have been proposed for Golgi (*Marr, 1969*; *Mauk and Donegan, 1997*) and candelabrum cells (*Osorno et al., 2022*), proposals regarding the functions of MLIs generally have been limited to the relatively simplistic notion that they contribute to feedforward inhibition of PCs (*Kim and Augustine, 2021*). Moreover, the function of stellate and basket cells generally is not considered separately in theories of cerebellar cortex.

Following up on suggestions that MLIs may receive inhibitory input from PCs (*O'Donoghue et al., 1989*; *Witter et al., 2016*), we present evidence from optogenetic mapping of local circuits and

paired recordings in cerebellar slices that some MLIs in the inner third of the molecular layer, but not stellate cells in the outer molecular layer, are inhibited by feedback from PCs. Because these MLIs, in turn, inhibit other PCs, our results suggest that PCs influence each other through disinhibition of MLIs. Consistent with this disinhibitory circuit, in vivo a subset of MLIs fire strongly anti-phasically with, and usually after, PCs during eyelid conditioning. Finally, large-scale computer simulations predict that PC feedback inhibition of MLIs enhances the ability of the cerebellum to learn and to facilitate more efficient information storage.

## Results

### Photostimulation of PCs

We used mice that express the light-activated cation channel, ChR2, exclusively in PCs to enable selective photostimulation of PCs in vitro. These mice were obtained by crossing transgenic mice expressing Cre recombinase behind a PC-specific promoter (PCP2) (*Barski et al., 2000*) with another transgenic mouse line expressing ChR2-H134R behind a floxed stop cassette (*Madisen et al., 2012*). Similar results were obtained using a second PCP2-cre line (*Zhang et al., 2004*; *Witter et al., 2016*). In these mice, light pulses generate photocurrents sufficient to evoke action potentials in PCs (*Asrican et al., 2013*). We examined ChR2 expression by imaging the yellow fluorescent protein (YFP) fused to the ChR2; virtually all PCs strongly expressed ChR2 in their somata and dendrites, with no fluorescence evident in MLI somata within the molecular layer (*Figure 1a*). Whole-cell patch clamp recordings in cerebellar slices from these double transgenic mice indicated that photostimulation reliably evoked action potentials in every PC examined (20/20), but was incapable of evoking any action potentials in MLI (0/406).

### Properties of PC to MLI feedback

We characterized synaptic connectivity between PCs and MLIs in sagittal slices of the cerebellar vermis, based on previous anatomical studies indicating that PC axon collaterals are oriented in the sagittal plane (*Watt et al., 2009*; *Witter et al., 2016*). We started by using relatively large spots of blue light (~0.2 mm$^2$ area) to photostimulate a large number of PCs (*Figure 1b*, top), while measuring postsynaptic responses in MLIs. Inhibitory postsynaptic currents (IPSCs) were observed only in MLIs located in the inner one-third of the molecular layer, where basket cells are located (*Figure 1b*, center). IPSCs were abolished by GABA$_A$ receptor blockers, such as bicuculline and gabazine, but were unaffected by the glutamate receptor blocker kynurenic acid, indicating that the IPSCs are monosynaptic and mediated by GABA$_A$ receptors. Connectivity between PCs and MLIs was observed in 16% of the 371 MLIs examined in the inner third of the molecular layer, where basket cells are located, in mice aged 3 weeks or older. No connections were detected (*n*=35) between PCs and MLIs located in the outer two-thirds of the molecular layer, where stellate cells are located (*Figure 1b*, bottom). We refer to the subset of MLIs receiving PC inputs as 'PC-MLIs' (*Figure 1b*, middle). The somata of PC-MLI were 60±2.9 μm (mean ± SEM, *n*=59) above the PC layer and thus are not candelabrum cells, interneurons whose somata reside within the PC layer (*Lainé and Axelrad, 1994*; *Osorno et al., 2022*). Inhibitory feedback from PCs to PC-MLIs was also observed in the lateral cerebellum: a total of eight connections were detected among 34 recordings in the inner molecular layer of simplex, crus I, and crus II lobules. This 24% rate of connections compares favorably to the ~16% observed in 371 recordings in the vermis. Therefore, the PC-to-PC-MLI feedback circuit seems widely distributed throughout the cerebellum and may play a general role in cerebellar computation. To determine how feedback from PCs affects the activity of PC-MLI, we photostimulated PCs with brief light pulses (5 ms) while depolarizing PC-MLIs to evoke rapid action potential firing. Light-evoked synaptic input from PCs was capable of completely suppressing PC-MLI activity (*Figure 1c*).

We then used recordings from pairs of connected PCs and MLIs for detailed characterization of the inhibitory feedback circuit. Although recordings from connected pairs were rarely obtained (see below), seven successful experiments (six whole-cell recordings and one cell-attached recording from PCs) allowed us to directly define the functional attributes of this circuit. Single action potentials (simple spikes) evoked in presynaptic PCs induced IPSCs in postsynaptic MLIs (*Figure 1d*). IPSC latency was brief, with a mean of 1.15±0.02 ms (*Figure 1e*). IPSCs were quite variable in amplitude (*Figure 1f*, red), with a coefficient of variation (CV) of 0.39±0.08 and peak amplitude of 56.1±20.3 pA at –50 mV

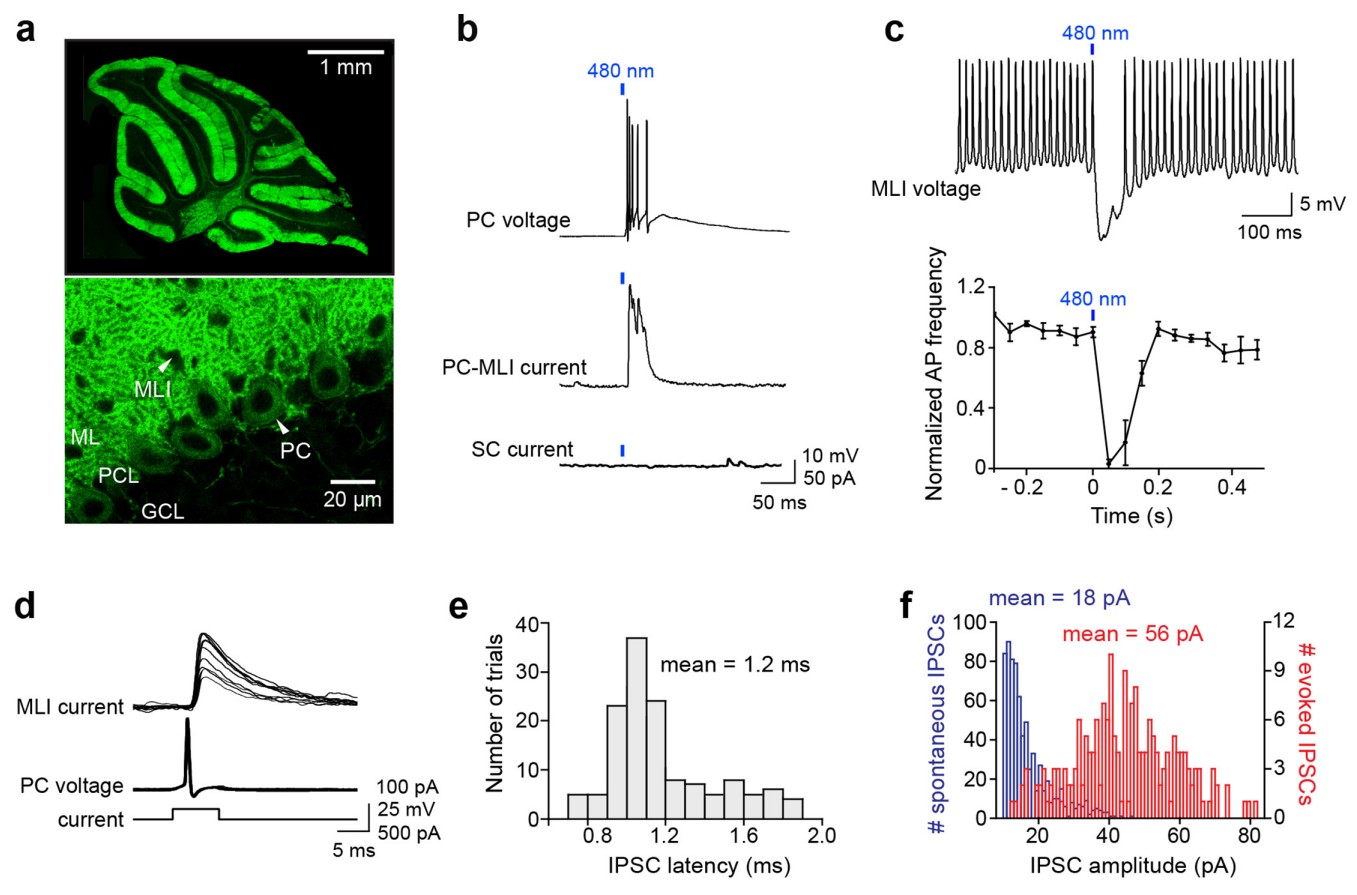

**Figure 1.** Optogenetic interrogation of Purkinje cell (PC) to molecular layer interneuron (MLI) circuit. (**a**) Selective expression of ChR2 in cerebellar PCs. Top: Image of ChR2-YFP fluorescence (green) in a sagittal cerebellum section from PCP2-Cre; Ai32 double transgenic mice. Strong ChR2 expression was observed throughout the entire cerebellar cortex, especially in the molecular layer where PC dendrites are located. Bottom: Higher-magnification image shows expression of ChR2-YFP in PC soma (arrowhead) and dendrites. Small black holes (arrowhead) in the molecular layer represent somata of MLI that do not express ChR2-YFP. ML, molecular layer; PCL, Purkinje cell layer; GCL, granule cell layer. (**b**) Brief illumination (480 nm, 5 ms, 9.9 mW/mm²) evoked action potentials in ChR2-expressing PCs (top). Photostimulated PCs induced inhibitory postsynaptic currents (IPSCs) in PC-MLI cells (center) but not in stellate cells (SC; bottom). (**c**) Top: Photostimulation of PCs inhibited firing in a postsynaptic MLI. PC-MLI was depolarized by a depolarizing current (20 pA) to sustain action potential firing. Bottom: Activation of PCs input (at blue bar) was sufficient to briefly but completely inhibit action potential firing in postsynaptic PC-MLI cells. Points indicate means and error bars represent SEMs (*n*=4). (**d**) Superimposed traces of recordings from a connected PC and PC-MLI pair. Action potentials in a presynaptic PC (middle traces), caused by depolarizing current pulses (bottom traces), induced IPSCs in the postsynaptic PC-MLI (top traces). PC-MLI holding potential was –50 mV. (**e**) Distribution of IPSC latencies measured in seven PC-MLI pairs (140 trials). Mean IPSC latency was 1.2 ms with relatively low synaptic jitter (0.03 ms). (**f**) Distribution of amplitudes of spontaneous (blue) and evoked (red) IPSCs in PC-MLIs.

The online version of this article includes the following source data for figure 1:

**Source data 1.** Source files for properties of inhibitory input from PCs to PC-MLIs.

(*n*=7). This indicates a synaptic conductance of 2.13±0.61 nS. The mean amplitude of spontaneous (presumed miniature) IPSCs was 18.0±1.3 pA (*Figure 1f*, blue), indicating a quantal content (ratio of amplitudes of evoked/spontaneous IPSCs) of 3.0±0.9 for evoked IPSCs. Recalculating based on IPSC integral, rather than peak amplitude, raised the estimate to 6.7; this reveals a substantial contribution of asynchronous GABA release (*Song and Augustine, 2016*). These unitary synaptic properties are similar to those reported for synapses formed between PCs by PC axon collaterals (*Orduz and Llano, 2007*; *Watt et al., 2009*; *Witter et al., 2016*). However, unlike PC-PC synapses (*Watt et al., 2009*; *Witter et al., 2016*), the PC-to-PC-MLI synapse exhibited very few failures (*Figure 1d*). This is a consequence of the relatively high quantal content and large readily releasable pool (RRP) of synaptic vesicles (99±16 quanta; *n*=15; see *Figure 2h* below). From the ratio of quantal content and RRP size, we estimate a release probability of 0.07.

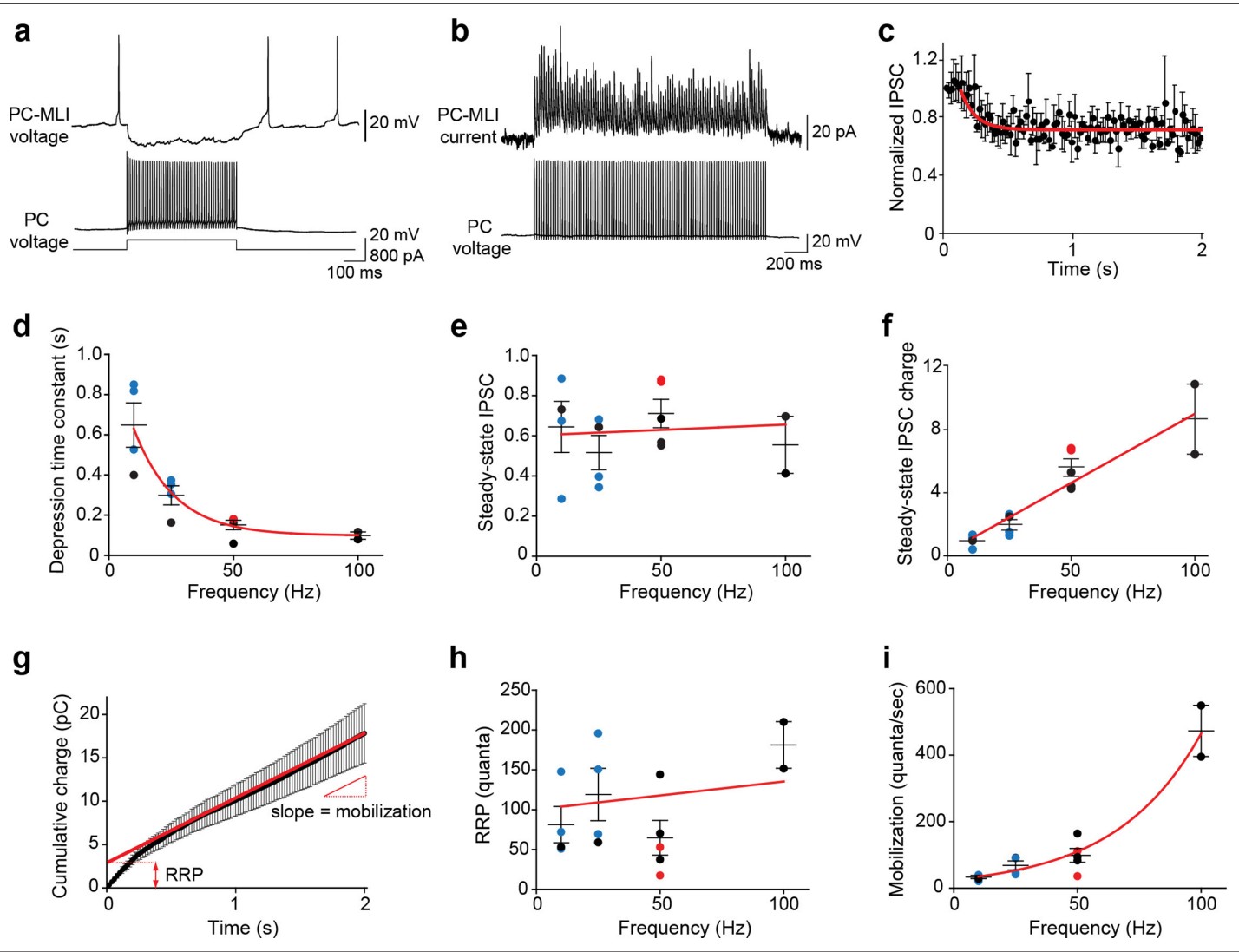

**Figure 2.** Frequency-independent depression at PC-to-PC-MLI synapses. (**a**) Synaptic transmission at a connected PC and PC-MLI pair. 100 Hz firing in the presynaptic PC (bottom) produced a sustained inhibition of the postsynaptic PC-MLI (top). Stimulus duration was 500 ms. (**b**) Average inhibitory postsynaptic currents (IPSCs) (10 trials) evoked by 100 stimuli (50 Hz, 2 s) from a connected PC and PC-MLI pair. During such trains of synaptic activity IPSCs often initially showed facilitation, indicated by an increase in IPSC amplitude, which was followed by relatively mild synaptic depression, indicated by a reduction in IPSC amplitude to levels below that of the first IPSC in the train. (**c**) Kinetics of synaptic depression during sustained synaptic activity. IPSC amplitudes were normalized to the first IPSC for each response in the 50 Hz stimulus train. Points indicate mean values and bars are ± SEMs (*n*=5), while red line is an exponential fit to the time course of depression of IPSCs. (**d**) Depression time constants were measured from exponential fits, as in (**c**), at different stimulation frequencies. Symbols represent data collected via different methods of stimulation of presynaptic PCs: blue = photostimulation; red = intracellular current injection in dual recordings; black = extracellular stimulation. Red line is an exponential fit to the data. (**e**) Steady-state amplitudes of IPSCs measured during stimulus trains of different frequencies; values are normalized to the first response in a train, as in (**c**). Red line indicates a linear fit to the data; slope of the line was not significantly different from 0 (p=0.85, F-test), indicating no significant influence of stimulus frequency on steady-state IPSC amplitude. (**f**) Steady-state charge transfer (IPSC amplitude × frequency) measured at different frequencies of synaptic activity. Colored symbols represent mean values determined for individual experiments, using the color code described in (**d**), and bars indicate ± SEM. Red line represents a linear fit to the data and indicates a dramatic, frequency-dependent enhancement of synaptic transmission. (**g**) Kinetics of synaptic charge transfer during a stimulus train. Points indicate IPSC charge integrated over time during 50 Hz stimulation, while bars indicate means ± SEM (*n*=5). The last 20 data points were fit by linear regression (red line) to estimate the size of the readily releasable pool (RRP; y-intercept) and mobilization rate (slope). (**h**) The analysis shown in (**g**) was used to measure RRP size at different stimulus frequencies. Red line indicates a linear fit to the data; slope of approximately zero indicates no consistent effect of stimulus frequency on RRP size. (**i**) Rate of synaptic vesicle mobilization from the reserve pool to the RRP was estimated using the analysis shown in (**g**). Points represent mean values determined for individual experiments, while bars are means ± SEM. Red line is an exponential fit to the data and indicates a large influence of stimulus frequency on the rate of vesicle mobilization. MLI, molecular layer interneuron; PC, Purkinje cell.

*Figure 2 continued on next page*

*Figure 2 continued*

The online version of this article includes the following source data and figure supplement(s) for figure 2:

**Source data 1.** Source files for properties of synaptic depression at PC-to-PC-MLI synapses.

**Figure supplement 1.** Synaptic transmission between a Purkinje cell (PC) and a PC-MLI at different frequencies of activation.

Because PCs fire repetitively at rates up to 100 Hz in vivo (*Thach, 1968*; *Zhou et al., 2014*), we determined how sustained activity affected transmission at the PC-to-PC-MLI synapse at near-physiological temperature (34–35°C). Trains of presynaptic PC activity (10–100 Hz) initially caused synaptic facilitation, which was followed by a mild depression of IPSC amplitude (*Figure 2a–b* and *Figure 2—figure supplement 1*). Depression followed an exponential time course, with the rate of depression depending upon stimulus frequency (*Figure 2c and d*). Remarkably, IPSC amplitude was sustained at approximately half of its initial strength even during prolonged activity, independent of stimulus frequency (*Figure 2e*). As a result, the steady-state amount of synaptic transmission – measured as total IPSC charge – paradoxically increased at higher frequencies (*Figure 2f*). By measuring cumulative synaptic charge during repetitive activity (*Figure 2g*), we determined the size of the RRP (*Figure 2h*) and the rate of mobilization of synaptic vesicles from a reserve pool (*Figure 2i*). This analysis revealed that while RRP size is independent of the rate of synaptic activity, PCs sustain their transmission via an activity-dependent acceleration of synaptic vesicle mobilization from the reserve pool to the RRP. This is similar to the behavior of synapses between PCs and deep cerebellar nuclear neurons (*Turecek et al., 2016*). Thus, PCs can inhibit PC-MLIs even during high-frequency activity, thereby maintaining an inverse relationship between the activities of these two neuron types.

## Mapping the spatial organization of the PC-MLI circuit

The circuit between presynaptic PCs and postsynaptic MLIs was mapped by scanning small spots of laser light (405 nm, 4 ms duration, ~1 µm diameter in the focal plane) to evoke action potentials in presynaptic PCs expressing ChR2 (*Wang et al., 2007*; *Kim et al., 2014*). Under these conditions, 1.2 µW or more of laser power could reliably evoke action potentials in PCs in cerebellar slices from mice 3 weeks or older. Optimal spatially resolved photostimulation (*Kim et al., 2014*) occurred at a laser power of 3 µW: at this laser power, light spots were capable of evoking action potentials at any location on the dendrite or cell body of individual PCs (*Figure 3a*, position 2), aside from their axons. In contrast, positioning the light spot more than 30 µm away from the PC soma along the PC layer failed to evoke action potentials (*Figure 3a*, position 1 or 3).

The spatial organization of the circuit was mapped by scanning the laser light spot to evoke action potentials in presynaptic PCs, while recording IPSCs in MLIs. A light-evoked IPSC indicated a synaptic connection between a PC photostimulated at that location and postsynaptic PC-MLI; thus, correlating light spot location with the amplitude of IPSCs mapped the position of presynaptic PCs (*Wang et al., 2007*; *Kim et al., 2014*; *Figure 3b*). Recordings from stellate cells in the outer molecular layer yielded blank maps (*Figure 3c*), again indicating an absence of connections between PCs and stellate cells (*n*=35). IPSCs were evoked in PC-MLI only when the light spot was positioned within the molecular and PC layers, where PCs are located. Presynaptic PC inputs to PC-MLI were most often revealed (59%) as a single cluster of pixels similar in size to the light-sensitive area of a single PC (*Figure 3d1*), occasionally (12%) as two discrete input areas (*Figure 3d2*) or a relatively large contiguous input area (29%; *Figure 3d3*). Within an input field, there were no obvious 'hot spots' of synaptic input, consistent with the homogeneous light sensitivity of individual presynaptic PCs (*Figure 3a*).

To estimate the number of PC inputs converging on a PC-MLI, we compared the area of the synaptic input field (*Figure 3d*) with the area over which light spots could evoke action potentials in single PCs (optical footprint, *Kim et al., 2014*; *Figure 3a*). Mean input field area was 1.5 times larger than the mean area of PC optical footprints, suggesting that on average a PC-MLI receives inhibitory input from only one or two converging PCs (*Figure 3e*). This is consistent with the observed spatial organization of input maps shown in *Figure 3d*. The spatial organization of the PC-to-PC-MLI circuit is different from that of the MLI-to-PC circuit (*Kim et al., 2014*): presynaptic PCs were largely found within 200 µm of their postsynaptic PC-MLI (*Figure 3f*) and were asymmetrically distributed, with most presynaptic inputs originating from either juxtaposed PCs or PCs located toward the apex of the

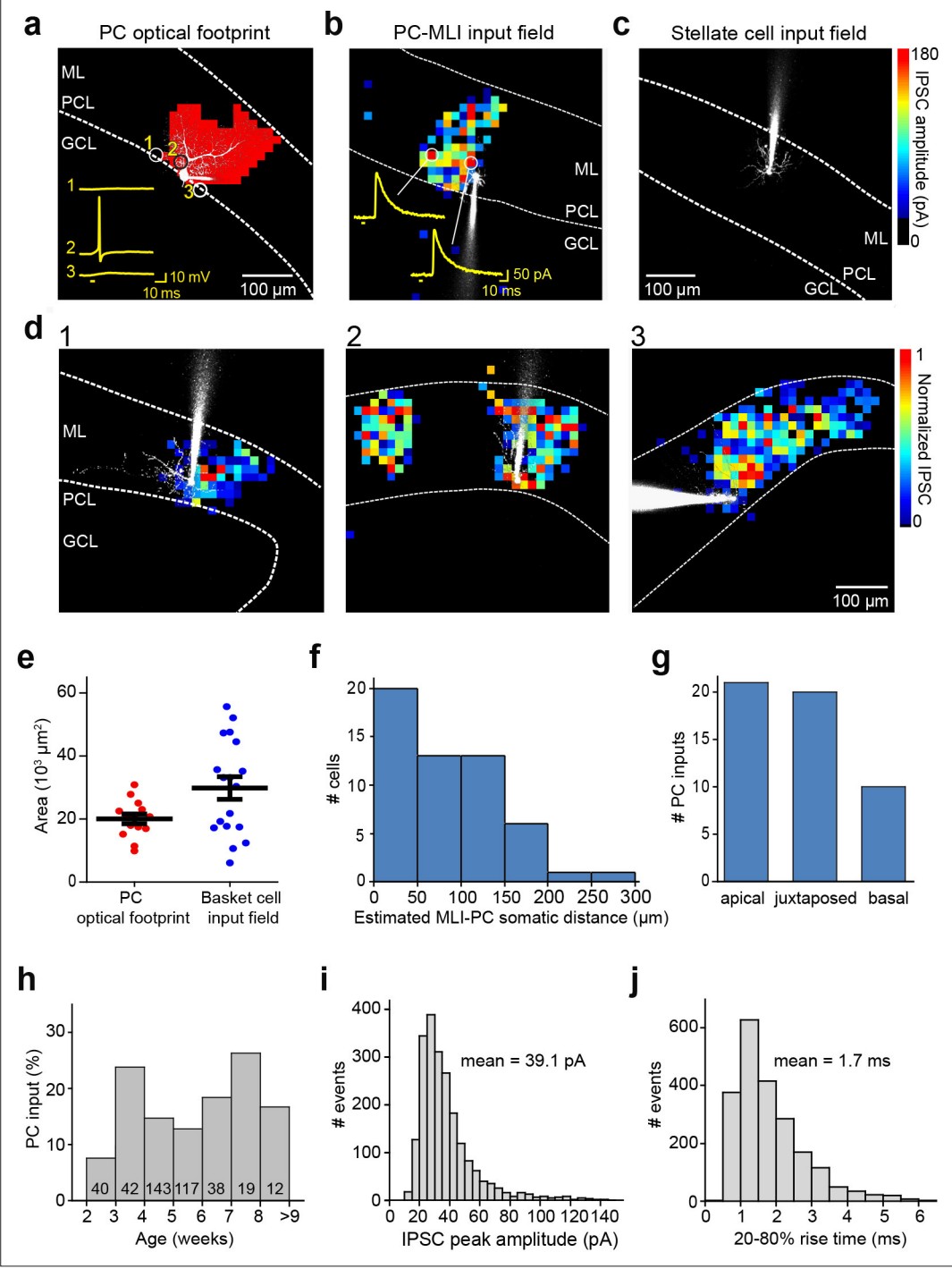

**Figure 3.** Spatial organization of PC-to-MLI circuit. (**a**) Scanning a brief laser light spot (405 nm, 3 μW, 4 ms duration) across a cerebellar slice revealed locations (red pixels) where photostimulation evoked action potentials in a ChR2-expressing, dye-filled PC (white). Numbered traces (yellow) in the inset show membrane potential changes produced when laser spot was located at the indicated pixels. Bar below traces indicates time of light flashes. Action potentials were evoked when the light spot was located over the cell body and entire dendritic region of the PC. ML, molecular layer; PCL, Purkinje cell layer; GCL, granule cell layer. (**b**) Map of inhibitory input from a presynaptic PC to a postsynaptic PC-MLI was created by correlating light spot location with amplitude of inhibitory postsynaptic currents (IPSCs) evoked by focal photostimulation of PCs. IPSC amplitudes are encoded in the pseudocolor scale shown at right. Traces show IPSCs evoked at the indicated locations and bars indicate time of light flashes. (**c**) No inhibitory input from PCs was detected in stellate cells. (**d**) Varied spatial organization of PC

*Figure 3 continued on next page*

*Figure 3 continued*

inputs to PC-MLIs. Maps illustrate IPSC amplitudes (pseudocolor scale at right) evoked in three different dye-filled PC-MLIs (white) shown in panels 1–3. (**e**) Comparison of input field area of PC-MLIs (blue, *n*=18) with PC optical footprints (red, *n*=14). (**f**) Distance between postsynaptic PC-MLIs and presynaptic PCs. PC soma location was estimated from the center of the input field, projected down to the PCL. (**g**) Orientation of postsynaptic PC-MLIs, relative to presynaptic PCs. (**h**) Connectivity between PCs and PC-MLIs measured at different ages. Numbers inside bars indicate sample sizes. (**i**) Distribution of amplitudes of IPSCs evoked by focal photostimulation of PCs. Holding potential was –40 mV. (**j**) Distribution of rise times of light-evoked IPSCs. MLI, molecular layer interneuron; PC, Purkinje cell.

The online version of this article includes the following source data for figure 3:

**Source data 1.** Source files for spatiotemporal organization of PC-to-PC-MLI circuit revealed by optogenetic mapping experiment.

lobule (*Figure 3g*). These observations are consistent with the anatomy of PC axon collaterals (*Watt et al., 2009*; *Witter et al., 2016*).

Synaptic inputs from PCs to PC-MLI could be detected as early as postnatal day 14 (P14; *Figure 3h*), a time when MLIs are still developing (*Altman, 1972*; *Pouzat and Hestrin, 1997*). However, the rate of connectivity at ages earlier than P21 (7.5%) was somewhat lower than at later ages (16%), suggesting that the circuit is established during the time that MLIs mature. After P21, the probability of detecting functional connections between PCs and PC-MLI was relatively constant up to P77 (*Figure 3h*), well after MLIs have fully matured (*Altman, 1972*; *Pouzat and Hestrin, 1997*). Thus, this synaptic circuit is functional into adulthood. No feedback inhibitory connections were observed between PCs and stellate cells at any time between postnatal weeks 3 and 9.

The large number of IPSCs collected during our mapping experiments enabled further characterization of synaptic transmission at the PC-to-MLI synapse. Within the PC input fields, IPSC amplitude varied substantially from pixel to pixel but this variability was not due to variations in photostimulation of presynaptic PCs (*Figure 3a*). The CV of light-evoked IPSCs was 0.28±0.02 in 18 experiments; this is similar to the CV of IPSCs observed in our paired recordings from PCs and PC-MLIs (*Figure 1d*) and provides a second indication of highly variable transmission at the PC-to-MLI synapse. Mean IPSC amplitude, measured at a holding potential of –40 mV, was 39.1±4.0 pA (*Figure 3i*; 2128 IPSCs in 18 recordings). This corresponds to a synaptic conductance of 1.15±0.12 nS, which is not significantly different from that measured in paired recordings (p=0.17, Welch's t-test). IPSC rise time (20–80%) was relatively fast (1.69±0.13 ms, *Figure 3j*), similar to IPSCs produced at PC-to-PC synapses (*Orduz and Llano, 2007*; *Watt et al., 2009*; *Witter et al., 2016*) but more rapid than the MLI-to-PC connection (5.6±0.5 ms time-to-peak; see Figure 5c of *Kim et al., 2014*).

## Non-reciprocal connections between PCs and MLIs

Given that MLIs provide feedforward inhibition to PCs, we next asked whether circuits between PCs and MLIs are reciprocal, that is, whether MLIs innervated by a given PC also inhibit the same PC. For this purpose, we combined paired electrophysiological recordings with optogenetic mapping. Due to the abundance of PCs, we used mapping to increase the probability of detecting functional PC-MLI connections: after finding a PC-MLI and identifying the location of its presynaptic PC by scanning the photostimulating laser spot, as in *Figure 3b and d*, recordings were made from the somata of PCs within the area where the light spot evoked IPSCs. Among 139 paired recordings made via this strategy, synaptic inputs from PCs to MLIs were detected in seven cell pairs. This 5% success rate is an underestimate of PC-to-PC-MLI connectivity; although our optogenetic mapping identified the volume in which presynaptic PCs were located, there was still a low likelihood of sampling the one or two presynaptic PCs within the dozens of PCs within this volume. The results from one successful experiment are shown in *Figure 4a*. Action potentials evoked in the PC by depolarizing current pulses elicited IPSCs in the postsynaptic PC-MLI (*Figure 4a*, upper traces). However, evoking action potentials in the PC-MLI did not elicit IPSCs in the PC, indicating that this pair did not have a reciprocal connection (*Figure 4a*, lower traces). This was the case for all coupled PC/PC-MLI pairs.

As an alternative strategy, we randomly selected PCs within 100 µm of an MLI located in the inner third of the molecular layer. Synaptic inputs from MLIs to PCs could be detected as IPSCs evoked in PCs in response to MLI stimulation (*Figure 4b*, lower traces, 13/54 pairs). However, in such

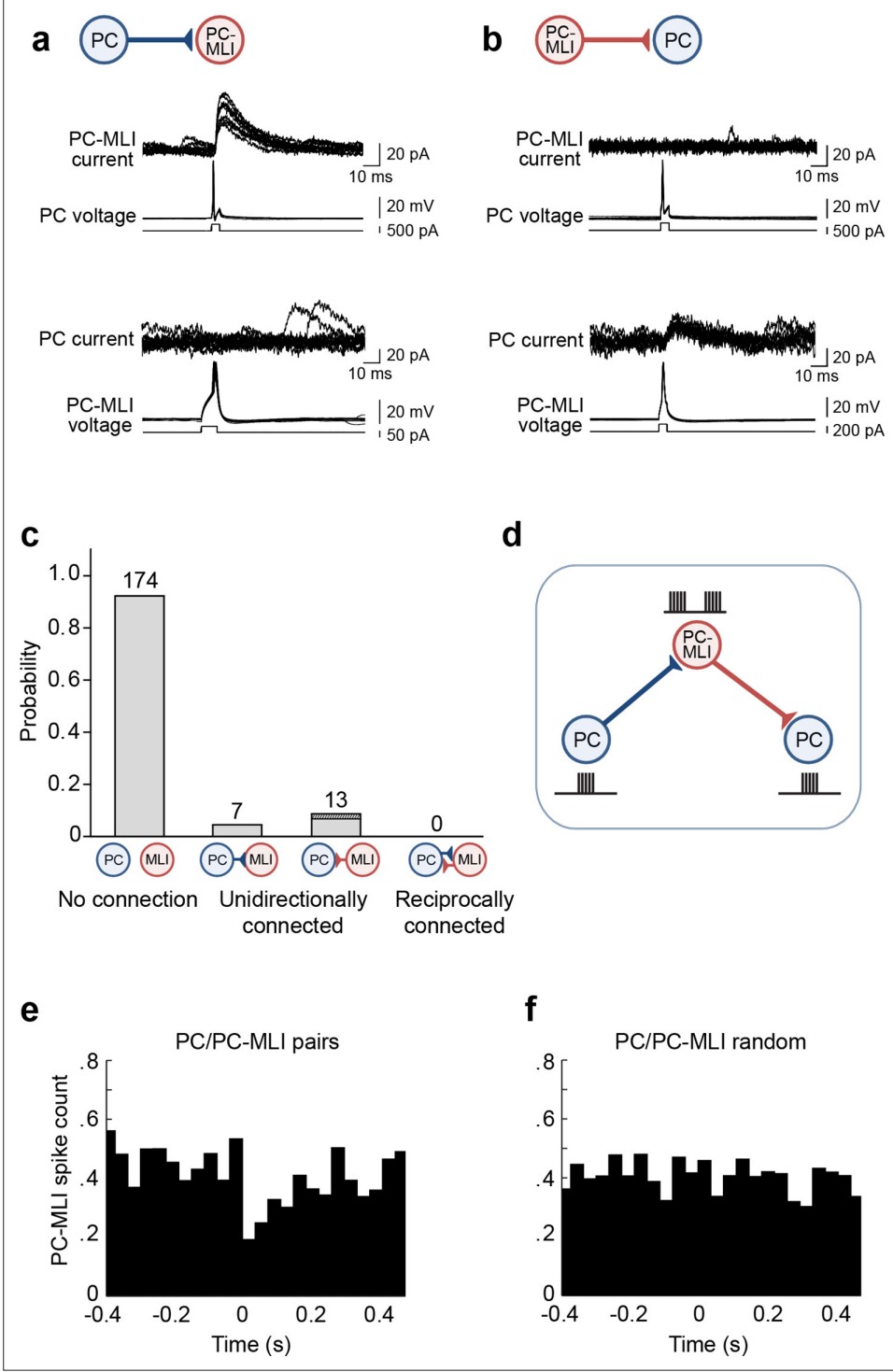

**Figure 4.** Evidence for non-reciprocal arrangement of inhibitory circuits between molecular layer interneurons (MLIs) and Purkinje cells (PCs). (**a**) Action potentials evoked in a presynaptic PC induced inhibitory postsynaptic currents (IPSCs) in a postsynaptic PC-MLI (top). However, action potentials in the PC-MLI did not evoke IPSCs in the PC (bottom). (**b**) Inhibitory inputs from PC-MLIs to PCs could also be observed (bottom). Again, these were non-reciprocal because stimulating the PC did not evoke IPSCs in the PC-MLI (top). (**c**) Probability of connections between PCs and MLIs observed in dual recordings. Only unidirectional connections were detected (*n*=194). Among the 13 MLI-to-PC connections observed, 3 (dark shading) involved confirmed PC-MLIs (as indicated by their inhibitory responses to PC photostimulation). (**d**) Proposed organization of circuits between PCs and PC-MLIs. PC-

*Figure 4 continued on next page*

*Figure 4 continued*

mediated inhibition of PC-MLI activity may disinhibit neighboring PCs, as indicated by schematics of activity next to each cell. (**e, f**) Temporal relationship between action potential firing, including both evoked and spontaneous PC action potentials, in connected PC/PC-MLI pairs in brain slices. Cross-correlations (at time lag = 0) between presynaptic PC and postsynaptic PC-MLI pairs (**e**) and activity in the same pair of cells after randomly shuffling the timing of PC spikes (**f**). The inhibitory connection between PCs and PC-MLIs yielded a strong, but transient, inverse activity correlation.

The online version of this article includes the following source data for figure 4:

**Source data 1.** Source file for temporal relationship in connected PC/PC-MLI pairs.

cases, action potentials in the postsynaptic PC invariably failed to evoke reciprocal IPSCs in the MLI (*Figure 4b*, upper traces). In 13 experiments where stimulation of an MLI in the inner one-third of the molecular layer evoked IPSCs in a PC, in no case did stimulation of the PC produce IPSCs in the MLI partner (*Figure 4c*). This includes three MLIs that were confirmed to be PC-MLIs, because they exhibited IPSCs in response to PC photostimulation. Overall, in a total of 194 paired recordings, only unidirectional connections (either PC-to-MLI or MLI-to-PC) or unconnected pairs were detected; in no case were reciprocally connected PCs and MLIs observed (*Figure 4c*). Although we cannot completely exclude the possibility of reciprocal connections, due to the limited number of recordings from connected pairs, our findings of no reciprocally connected cells in 10 connected PC/PC-MLI pairs indicate it is unlikely that these connections are reciprocal. Simulations indicate that our results would almost never be obtained (probability less than 0.001) if 50% of PC/PC-MLI connections were reciprocal, a 0.05 probability of observing no reciprocal connections in 10 trials if 20% of all connections are reciprocal and a probability greater than 0.9 if 1% or less of the connections are reciprocal. In summary, our data indicate that few, if any, PC/PC-MLI connections are reciprocal; thus, PCs will influence other PCs primarily through disinhibition, while not influencing their own behavior through recurrent inhibition (*Figure 4d*).

To describe how an action potential in a presynaptic PC influences spiking in postsynaptic PC-MLIs, we examined the temporal relationship between spikes in PCs and PC-MLIs (four cell pairs). This cross-correlation analysis revealed that inhibitory inputs from PCs to PC-MLIs produced a clear inverse correlation in the activity of these two cell types (*Figure 4e*), similar to the results shown in *Figure 1c*. Shuffling the timing of PC spikes eliminated this effect (*Figure 4f*), indicating that the inverse correlation normally observed after a PC action potential is genuine and is not an artifact of our analysis method.

## Cerebellar processing suggested by circuit simulation

To determine the role of this feedback circuit in cerebellar network function and behavior, we examined the activity of PCs and MLIs during eyelid conditioning, a cerebellar-dependent form of motor learning (*Halverson et al., 2015*). Eyelid conditioning involves pairing of eyelid stimulation, as an unconditioned stimulus (US), with a preceding tone that serves as a conditioned stimulus (CS). During repeated pairing of the CS with the US, the eyelid gradually produces a conditioned response (CR) to the CS. This paradigm allowed us to examine the relationship between PC and PC-MLI activity during a well-characterized and cerebellum-dependent behavioral task, both in silico and in vivo.

To provide a controlled experimental platform to predict how PCs and PC-MLIs interact in vivo – in the presence and absence of feedback inhibition from PCs to PC-MLIs (red in *Figure 5a*) – we used a previously established large-scale computer simulation (*Medina and Mauk, 1999*; *Medina et al., 2000*; *Medina et al., 2001*; *Medina et al., 2002*; *Ohyama et al., 2010*; *Kalmbach et al., 2011*; *Khilkevich et al., 2018*). The details of this simulation are provided in the Methods section. A variety of interstimulus intervals (ISIs) were used for training the simulation, to yield CRs that varied widely in their timing. After training, the activity of MLIs (stellate and basket cells) was not correlated with CRs in the absence of feedback inhibition (*Figure 5b1*). However, providing PC feedback inhibition to a subset of MLIs, to match the PC-to-PC-MLI connectivity observed in our slice experiments, yielded a bimodal distribution of correlations: a subpopulation of MLIs had activity that was highly correlated with eyelid CRs (mean $r=0.60 \pm 0.06$ SEM), while the activity of other MLIs had a minimal correlation with CRs (mean $r=-0.02 \pm 0.01$ SEM) (*Figure 5b2*). Thus, feedback inhibition causes PC-MLI activity to be time-locked to cerebellar output (i.e. eyelid CRs).

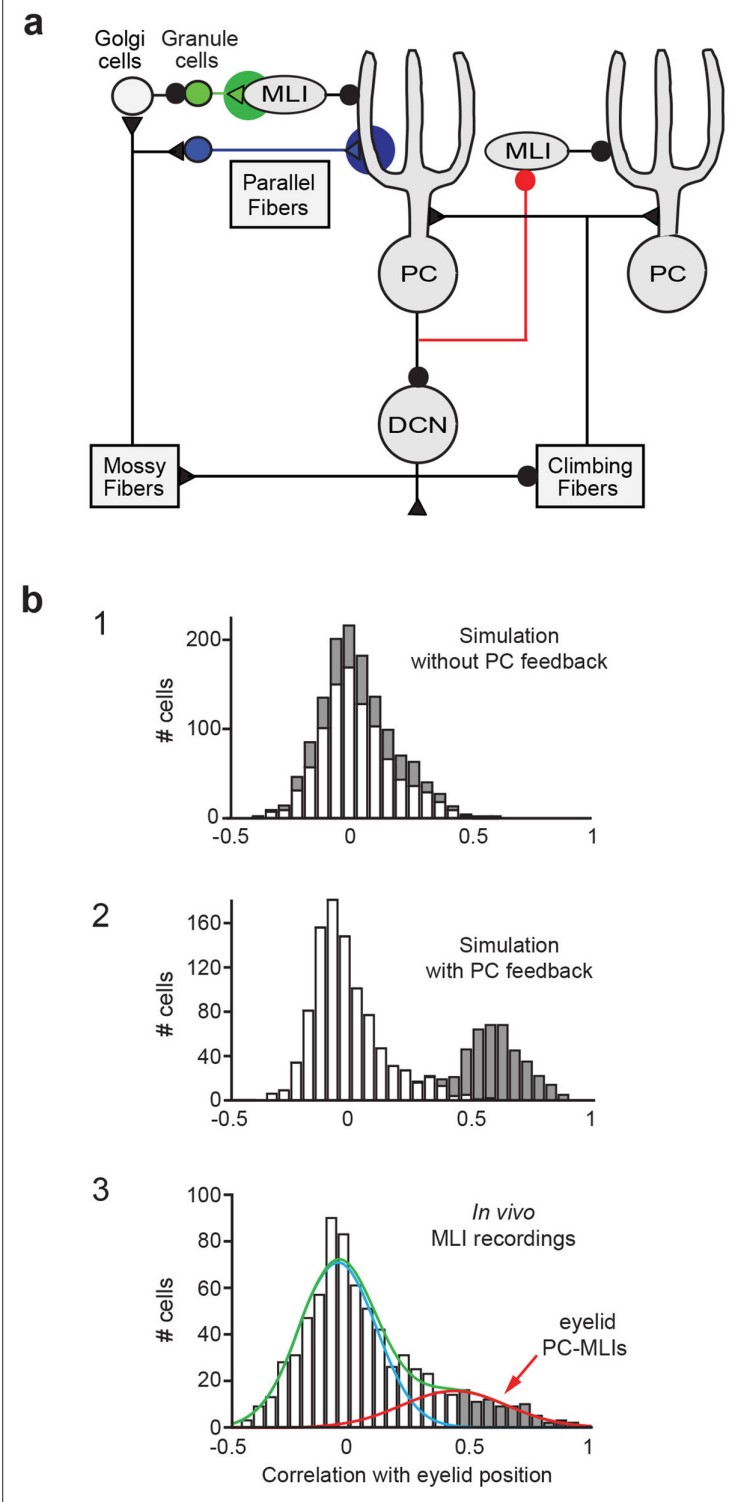

**Figure 5.** Relationship between molecular layer interneuron (MLI) activity and eyelid conditioning behavior.
(**a**) Circuit diagram showing main connections and sites of plasticity for the computer simulation of the cerebellum.
Green indicates potential plasticity at granule cell-to-MLI synapses and blue indicates plasticity at granule
cell-to-PC synapses. Red indicates the inhibitory feedback connection from PCs to MLIs that is the main focus
of our paper. (**b**) Cross-correlations between MLI activity and eyelid conditioned responses (CRs). The top two
panels show data from a computer simulation of the cerebellum. In well-trained simulations, correlations were
calculated between MLI activity and the predicted CRs. (1) The simulation without feedback from PCs to MLIs

*Figure 5 continued on next page*

*Figure 5 continued*

yielded a distribution of correlations centered around zero for all MLIs (white bars indicate stellate cells and gray bars indicate basket cells). (2) Simulations with PC-to-MLI feedback exhibited two distributions, one from a subset of MLIs that is shifted in the positive direction (gray bars) and the other centered around zero (white bars). (3) Distribution of correlations between CRs and MLI activity recorded in vivo for all interstimulus intervals (ISIs) can be fit with two distributions, one shifted in the positive direction (red curve) and one centered around zero (blue curve) similar to the simulation with the PC to MLI feedback. The cutoff for categorizing an MLI as a PC-MLI (gray bars) was a correlation coefficient ≥0.43, which represents the point where there was a 0.95 probability that the two fitted distributions were different. PC, Purkinje cell.

The simulation provided three experimentally testable predictions regarding the activity of PCs and PC-MLIs during eyelid conditioning: (1) There should be a bimodal distribution of correlations between MLI activity and behavioral CRs, with the highly correlated group being PC-MLIs (as in *Figure 5b2*). Thus, MLIs whose in vivo activity is correlated with CRs are likely to be PC-MLIs. (2) Because the activity of a subset of PCs (called eyelid PCs) decreases during CRs (*Heiney et al., 2014*; *Halverson et al., 2015*), the simulation predicts that the PC-to-PC-MLI feedback circuit will cause the activity of PC-MLIs to increase (via disinhibition) as presynaptic eyelid PCs become less active during the CR. Further, if feedback inhibition of PC-MLIs is indeed responsible for the inverse activity of PCs and PC-MLIs, then the simulation also predicts that (3) decreases in PC activity should precede increases in MLI activity during CRs. The following sections describe experiments that test these predictions via in vivo recordings of cerebellar neuron activity during CRs.

## Activity of some MLIs is correlated with conditioned eyelid responses

To illustrate motor learning behavior, *Figure 6a1* depicts eyelid responses in rabbits after conditioning. In this and subsequent figures, the gray shaded area represents the timing of the CS (tone). Anticipatory eyelid movements occur in response to the CS only after conditioning, indicated by upward deflections (dark gray) during the CS. In *Figure 6a*, these movements are sorted according to the time of eyelid closure relative to the start of the CS, with CR onset latency indicated by red points. Simultaneous measurements of action potentials (simple spikes) in an eyelid PC, identified by the presence of US-evoked complex spikes (*Halverson et al., 2015*), during these CRs are shown in *Figure 6a2*. A peristimulus time histogram (*Figure 6a2*, upper) illustrates that the activity of this eyelid PC decreased during the CRs. A raster plot (*Figure 6a2*, lower) reveals that these decreases in eyelid PC activity were highly correlated, on a trial-by-trial basis, with the time of onset of eyelid movement (red points). Previous work has established that eyelid CRs are driven by these learning-dependent decreases in eyelid PC activity (*Heiney et al., 2014*; *Halverson et al., 2015*).

The simulation predicts that PC-MLI activity should be correlated with eyelid CRs. We tested this prediction by recording MLI activity during eyelid CRs. In extracellular recordings, MLI activity can be distinguished by an established algorithm that categorizes neurons according to their baseline firing properties (*Simpson et al., 2005*). We observed that a small subset of MLIs increased their activity at the time that CRs occurred. *Figure 6b2* shows an example of such an MLI: the activity of this cell increased during the CR, evident in a peristimulus time histogram (upper) and in raster plots (lower) of action potential firing. The corresponding eyelid CRs are shown in *Figure 6b1*. The additional examples shown in *Figure 6c* and *Figure 6—figure supplement 1b–d* also illustrate striking correlations of MLI activity with the time of CR onset. Because only a subset of MLIs receive PC feedback, the model predicts that not all MLI responses should be well correlated with CRs (*Figure 5b2*); consistent with this, *Figure 6c* shows examples of MLIs whose correlation with CRs ranges from 0.2 to 0.91.

We analyzed the correlation between MLI activity and CR timing during eyelid conditioning for a total of 814 cells (10 rabbits), using three different ISIs. As in the simulation, the majority of MLIs had little or no correlation with the kinematics of CRs (*Figure 5b3*). This broad range of MLI activity could be fit by the sum of two Gaussian distributions (blue and red curves in *Figure 5b3*). The means of each MLI subpopulation (red: $r=0.43 \pm 0.01$ SEM, blue: $r=-0.05 \pm 0.01$ SEM) were similar to those predicted by the simulations that included PC feedback (*Figure 5b2*). The skewness (rightward shift) of the in vivo distribution (0.84 in *Figure 5b3*) also more closely matched the skewness (0.65) of the simulated distribution with PC feedback (*Figure 5b2*) than that of the simulated distribution without PC collaterals (0.43; *Figure 5b1*). These results are consistent with prediction (1) above.

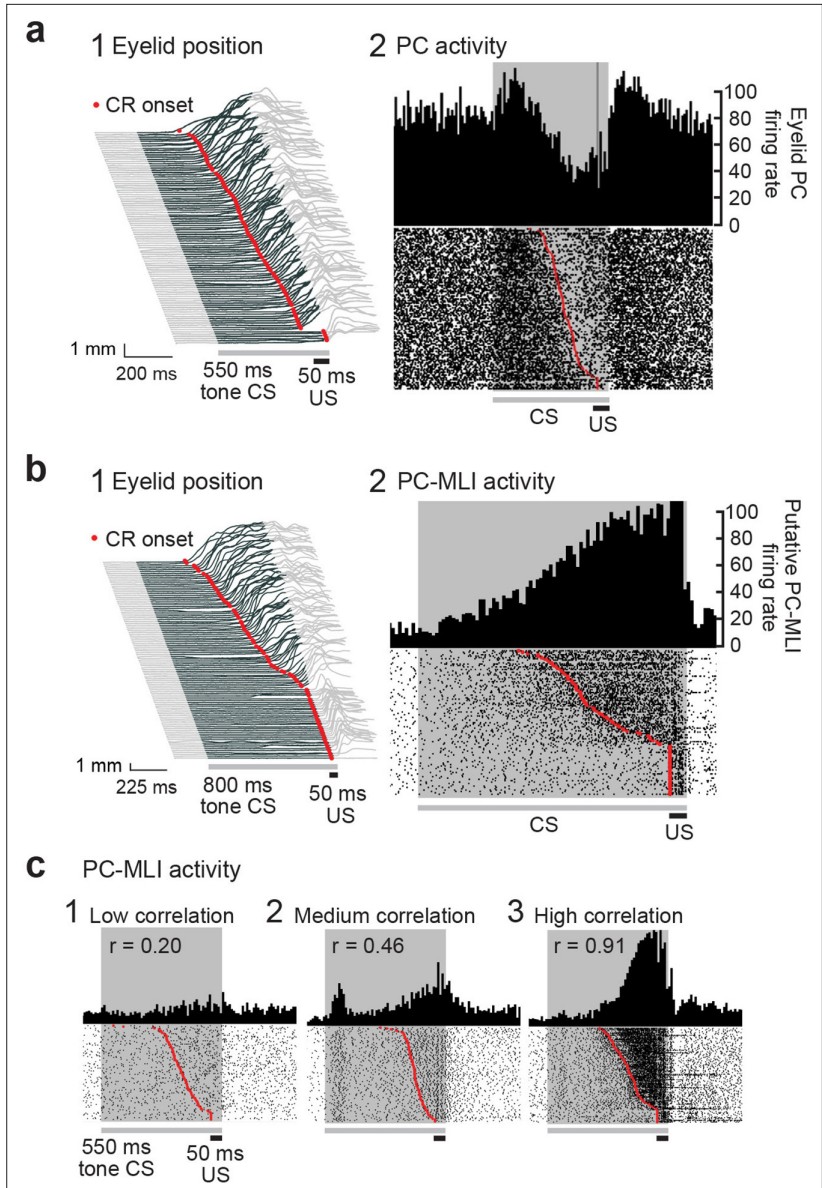

**Figure 6.** Changes in Purkinje cell (PC) and PC-MLI activity correlated with eyelid position during eyelid conditioning. (**a**) (1) Eyelid movements (vertical) in response to conditioned stimuli (CS; tone) and unconditioned stimuli (US; eyelid stimulation), sorted by the latency of the conditioned response (CR; red points). Upward deflection indicates eyelid closure, with closure indicated by dark gray lines reflecting CRs. (2) Peri-stimulus time histogram (top; 10 ms bins) and raster plot (bottom) of an eyelid PC recording from all trials of the eyelid conditioning session shown at left. For this and all subsequent raster plots, the gray bar under the plot indicates duration of the CS and the black bar indicates the duration of the US. (**b**) Peri-stimulus time histogram (top; 10 ms bins) and raster plot (bottom) of a recording from a putative PC-MLI (2), along with the behavior from all trials of the eyelid conditioning session during the recording (1), sorted by the latency of the CRs (red points). (**c**) Examples of the wide range of MLI responses observed in vivo during CRs. Peri-stimulus time histograms (10 ms bins) and raster plots sorted by response latency (red points) for three MLIs. Example 1 is from the low-correlation (*r*=0.20) part of the distribution shown in *Figure 5b3*. Examples 2 (*r*=0.46) and 3 (*r*=0.91) were operationally defined as eyelid PC-MLIs due to their high correlation with CR behavior. MLI, molecular layer interneuron.

The online version of this article includes the following figure supplement(s) for figure 6:

**Figure supplement 1.** Relationship between simultaneously recorded activity of eyelid molecular layer interneurons (MLIs) and eyelid Purkinje cells (PCs) and conditioned eyelid responses.

## Activity of putative PC-MLIs increases during conditioned eyelid responses

The subset of MLIs showing a relatively high correlation with CRs during eyelid conditioning are likely to be PC-MLIs; thus, we will refer to them as putative PC-MLIs. We operationally defined putative PC-MLIs as cells whose activity was correlated with eyelid position with a correlation coefficient of 0.43 or greater. This criterion identifies MLIs with a 95% or greater probability of being within the high-correlation population (red curve) of MLIs that create the high skewness of the distribution shown in *Figure 5b3*. Most of the 71 putative PC-MLIs identified in this way (8.7% of all MLIs) came from sessions where the activity of nearby eyelid PCs was recorded simultaneously, to ensure that the PC-MLIs were nearby the eyelid PCs (within the detection range of the recording tetrodes, which is approximately 120 μm). These connectivity rates and distances are comparable to our observations in cerebellar slices, where we found that approximately 5–6% of MLIs receive PC feedback inhibition (*Figure 1b*) that extends over 200 μm or less (*Figure 3*). Correlations between the activity of all MLIs examined with eyelid position and velocity are shown in *Figure 6—figure supplement 1a*.

Recordings from the 71 putative PC-MLIs – including the examples shown in *Figure 6b2, c2 and c3* – reveal two clear relationships between the activity of putative PC-MLIs and eyelid CRs: (i) there was little or no response in these neurons in trials where no CR occurred (bottom of the sorted raster plots) and (ii) there was a burst of activity in putative PC-MLIs that closely tracked the time of onset of CRs (red points) on a trial-by-trial basis. These results are consistent with prediction (2) of the simulation. In many MLIs – both putative PC-MLIs and other cells identified as MLIs – an additional increase in activity was often observed at the beginning of the CS. Examples of such responses are shown in *Figure 6c2 and c3*. Such responses were also observed in PCs and were not correlated with the occurrence of a CR; thus, they are sensory responses associated with detecting the sound stimulus.

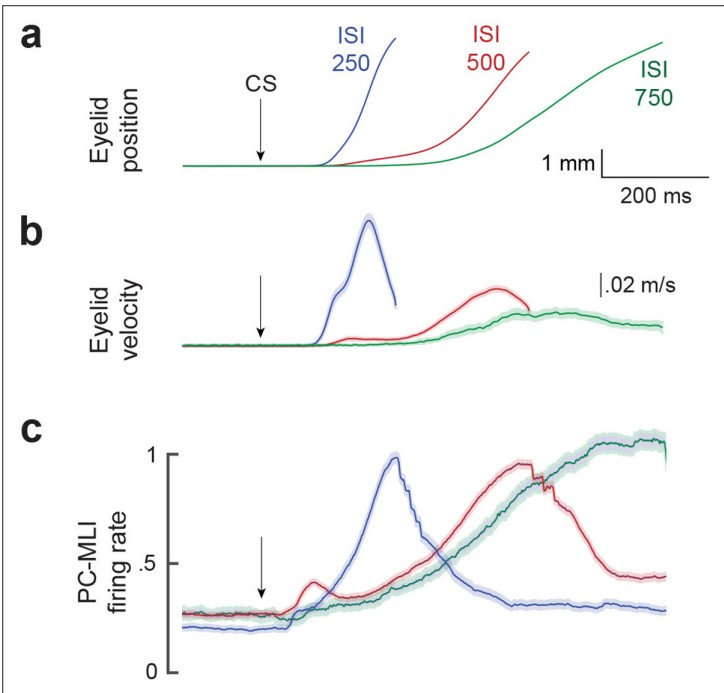

**Figure 7.** Comparison of mean firing rate of PC-MLI to the kinematics (position and velocity) of eyelid conditioned response (CR). Average eyelid position (**a**), velocity (**b**) and normalized average eyelid PC-MLI activity (**c**) shown for three different interstimulus intervals (ISIs) (ISI 250, 25 PC/PC-MLI pairs; ISI 500, 36 pairs; ISI 750, 10 pairs). The shaded region represents the 95% confidence intervals and arrows represent time of conditioned stimulus (CS) onset. MLI, molecular layer interneuron; PC, Purkinje cell.

The online version of this article includes the following source data and figure supplement(s) for figure 7:

**Figure supplement 1.** Normalized eyelid PC-MLI activity sorted by response onset or amplitude.

**Figure supplement 1—source data 1.** Source files for eyelid position and normalized eyelid PC-MLI firing rate sorted by conditioned response (CR) onset and amplitude (*Figure 7—figure supplement 1*).

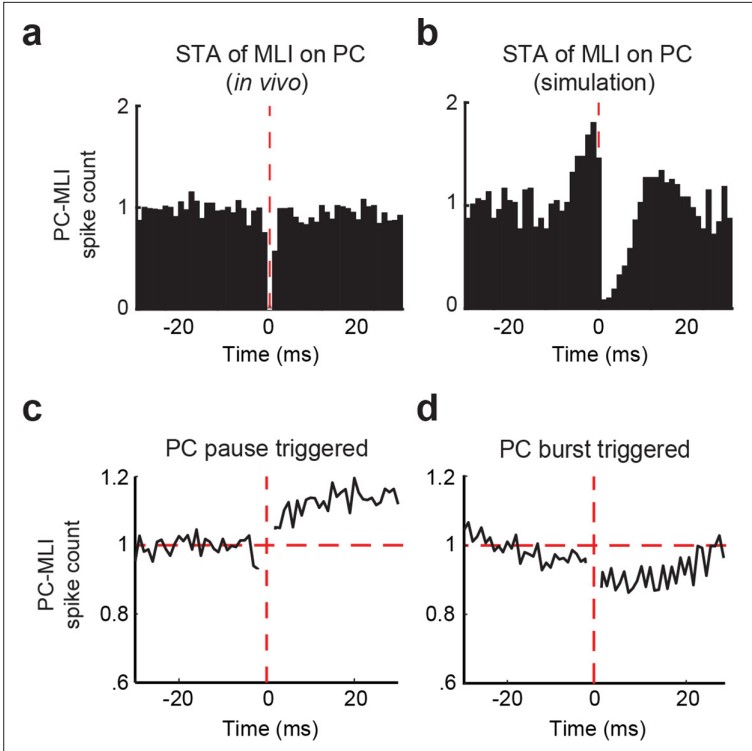

**Figure 8.** Temporal relationship between connected Purkinje cells (PCs) and putative PC-MLIs. (**a and b**) Average cross-correlograms calculated from significantly modulated pairs of MLIs and eyelid PCs. Background activity of putative PC-MLI is triggered off PC simple spikes (vertical red line), for PC-MLI and PC pairs recorded on the same tetrode (**a**) and from the computer simulation (**b**). Decrease in MLI activity after a PC simple spike indicates an inverse relationship between the activity of the two cell types. (**c and d**) Average PC pause-triggered (**c**) and burst-triggered (**d**) cross-correlograms of putative PC-MLI background activity. Plots show average PC-MLI activity across significant pairs recorded on the same tetrode. Pauses in PC simple spikes resulted in an increase in mean PC-MLI activity, while bursts of PC simple spikes resulted in a decrease in mean PC-MLI activity. MLIs, molecular layer interneurons.

The online version of this article includes the following figure supplement(s) for figure 8:

**Figure supplement 1.** Relationship between baseline activity of PC-MLIs and PCs simultaneously recorded from the same tetrode.

The activity of putative PC-MLIs was also correlated with the kinematics of conditioned eyelid movements. *Figure 7* compares the mean time course of the activity of all putative PC-MLIs (*Figure 7c*) to the position (*Figure 7a*) and velocity (*Figure 7b*) of eyelid CRs, sorted according to the ISI used for training. Despite the loss of temporal precision imposed by averaging, the firing rate of putative PC-MLIs corresponded well to both average eyelid position and velocity at each ISI. These high correlations are likely a result of activity driven by feedback from eyelid PCs, whose activity highly anti-correlates with both measures (*Halverson et al., 2015*). Additional analyses revealed that the activity of putative PC-MLIs closely tracked both the onset latency and amplitude of CRs for each ISI tested (*Figure 7—figure supplement 1*). In sum, we conclude that conditioning-related increases in PC-MLI activity are closely linked to eyelid CRs, consistent with prediction (2) of the simulation.

## Inverse relationship between activity of putative PC-MLIs and PCs

To test prediction (3) of the simulation, we examined the relationship between the activity of PCs and putative PC-MLIs during CRs. For this purpose, we employed recordings of the activity of pairs of PCs and putative PC-MLIs detected by the same tetrode. We began by considering the baseline activity for 10 s prior to CS onset and calculated cross-correlograms of putative PC-MLI activity triggered by PC simple spikes. We were unable to evaluate the data at time 0 because simultaneously recorded action potentials from two units on the same tetrode merge into a signal that cannot be assigned to

either unit during sorting (*Gao et al., 2012*). Following a PC spike, there was a significant reduction in putative PC-MLI firing in six PC/PC-MLI pairs (*Figure 8a* and *Figure 8—figure supplement 1a–b*). This relatively small and brief decrease in activity was specific to PC/PC-MLI pairs identified on the same tetrode. The pattern of cross-correlation between connected PCs and putative PC-MLIs was qualitatively similar to that observed in slices (*Figure 4e*) and was abolished when the timing of PC spikes was shuffled (*Figure 8—figure supplement 1a–b*).

We also performed a similar cross-correlation analysis on PC/PC-MLI pairs in the cerebellar simulation. There was a qualitative match between the in vivo recordings and simulations: a transient reduction in PC-MLI firing rate was observed following a PC action potential (*Figure 8b*) and was abolished when PC spike activity was randomized. A weaker effect of single PC spikes in vivo, compared to the effects observed in simulations (*Figure 8b*) and in slices (*Figure 4e*), probably arises from the much larger number of spontaneous excitatory inputs to each neuron in vivo. Differences in the timescale of the responses between in vivo and brain slice data are likely due to the lower neuronal firing rate in slices (which yields longer time intervals between PC spikes) as well as temperature differences between the two experimental systems. In summary, cross-correlation analysis from all three data sets demonstrated an inverse relationship between PC and PC-MLI activities, which is consistent with prediction (3). This correspondence strengthens the conclusions that PCs provide inhibitory feedback to PC-MLIs in vivo and that putative PC-MLIs identified in vivo are equivalent to the PC-MLIs identified in cerebellar slices.

Because of the relatively modest effects of individual PC action potentials on PC-MLI activity observed in vivo (*Figure 8a*), we extended our cross-correlation analysis to the larger bouts of PC activity occurring during action potential bursts, as well as the large reductions occurring during pauses in PC spiking. These analyses were restricted to recordings done between conditioning trials, to avoid the large changes in activity that occur during CRs (e.g. *Figure 6a and b* and below). Pauses were defined as the largest 35% of inter-spike intervals, while bursts were the smallest 35% of inter-spike intervals (50% for simulations). When spikes for a PC/PC-MLI pair were aligned to the start of a pause in PC activity, we found that these pauses clearly increased activity in putative PC-MLIs (*Figure 8c*; average of 11/27 PC-PC-MLI pairs detected on same tetrode). Further, bursts of PC activity decreased putative PC-MLI activity (*Figure 8d*; average of 13/27 PC-PC-MLI pairs). Examples of responses of individual pairs of PCs and PC-MLIs during PC pauses and bursts also showed strongly anti-correlated activity (*Figure 8—figure supplement 1c–d*).

These results indicate that the inverse relationship between PC and PC-MLI activity observed in vivo is more prominent during large changes in PC activity, compared to single PC spikes. The need for relatively large changes in PC activity in vivo highlights the importance of the frequency-independent synaptic transmission at the PC-to-PC-MLI synapse illustrated in *Figure 2*. By demonstrating that the inverse relationship between PC and PC-MLI activity is driven by changes in PC firing, these results are consistent with prediction (3) of our simulation.

We further tested prediction (3) by simultaneously measuring the activity of PCs and putative PC-MLIs during eyelid conditioning at three different ISIs. Examples of such recordings are shown in *Figure 9*. Peristimulus time histograms for the paired activity of an eyelid PC (green) and a putative PC-MLI (black) are shown in *Figure 9a and b*. In each case the eyelid PC and the putative PC-MLI fired inversely, with decreases in PC activity and increases in putative PC-MLI activity both corresponding – on a trial-by-trial basis – with the eyelid CR (red points). For all three examples in *Figure 9*, eyelid PC activity decreases before both CR onset and the increase in putative PC-MLI activity on every trial when a CR occurred; this is clearest in superimposed plots of average firing rate of both cells (*Figure 9c*). This temporal pattern is blurred when calculating the average firing rate for all eyelid PCs and putative PC-MLIs (*Figure 9—figure supplement 1a*) highlighting the importance of investigating individual pairs to determine the sequence of firing of PCs and putative PC-MLIs. In general, eyelid PC activity not only declined faster than PC-MLI activity increased, but also reached a minimum earlier during the CS. These unambiguous results are consistent with prediction (3) of the simulation.

Because of the complex kinetics of the responses of eyelid PCs and PC-MLI during CRs, we employed two alternative approaches to quantify the precise temporal relationship between these two responses. At the single-trial level, we measured the delay between the time at which PC activity reached 50% of its minimum during the CS and the time at which the activity of putative PC-MLIs reached 50% of its maximum for the pairs shown in *Figure 9*. For each ISI, whenever a CR was present

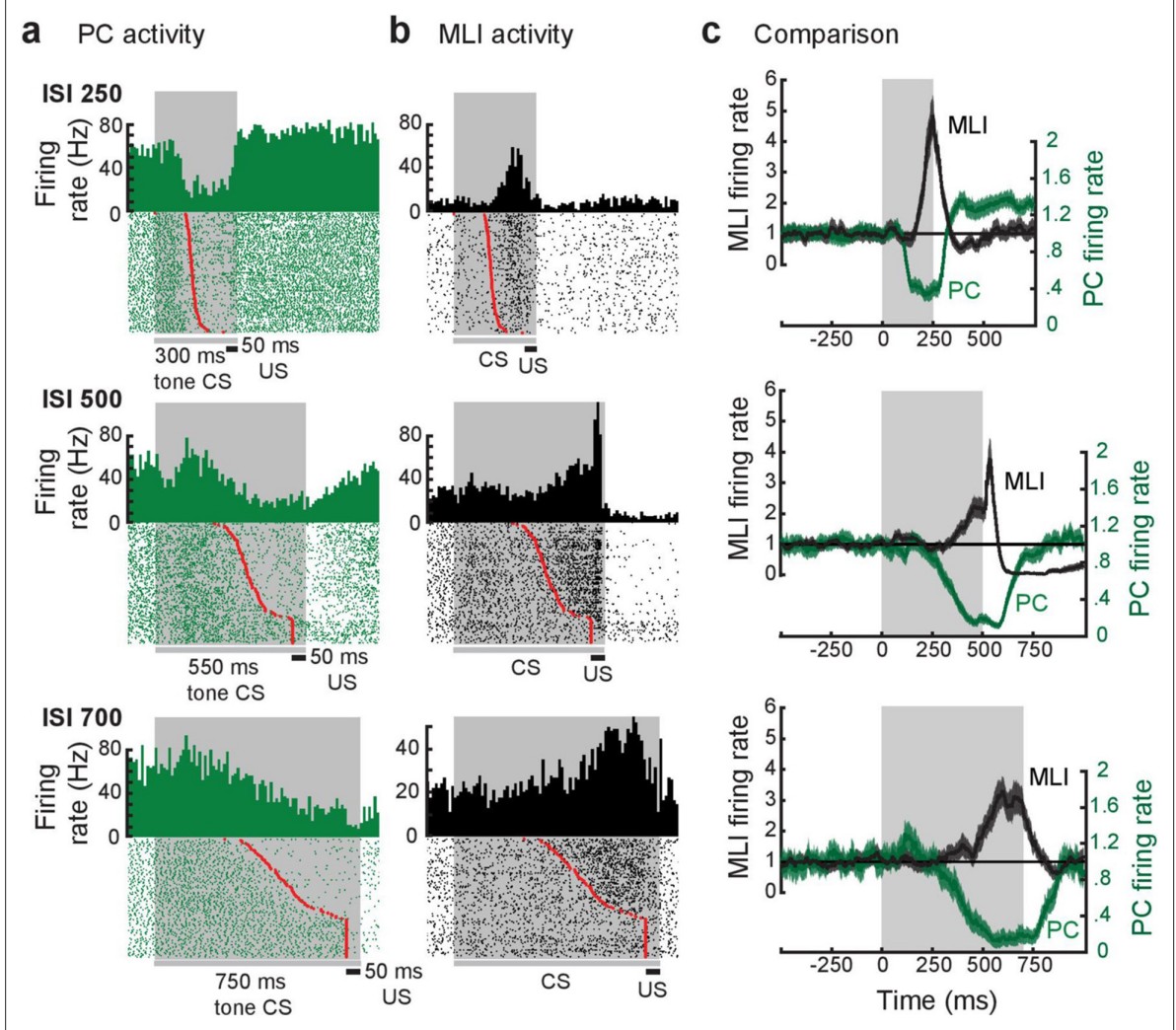

**Figure 9.** Relationship between individual simultaneously recorded eyelid molecular layer interneurons (MLIs) and eyelid Purkinje cells (PCs) during expression of conditioned eyelid responses. (**a and b**) Peri-stimulus time histograms (10 ms bins) and raster plots sorted by eyelid conditioned response (CR) latency (red points) of the simultaneously recorded activity of eyelid PCs and MLIs. The dark gray bars and shaded areas indicate the duration of the tone stimulus (conditioned stimulus [CS]) and the black bars indicate the duration of the eyelid stimulation (unconditioned stimulus [US]). (**c**) Normalized instantaneous firing rate for the examples in **a** and **b**. For all three examples, the activity of the eyelid PC decreases before the simultaneously recorded activity of the eyelid PC-MLI increases, as well as prior to the time of onset of the conditioned response. The gray shaded region indicates the interstimulus interval (ISI) for each pair.

The online version of this article includes the following source data and figure supplement(s) for figure 9:

**Figure supplement 1.** Relationship between activity of pairs of eyelid Purkinje cells (PCs) and putative PC-MLIs.

**Figure supplement 1—source data 1.** Source files for the time difference (delay time; MLI – PC) reaching 50% maximum response between eyelid PC and eyelid PC-MLI (*Figure 9—figure supplement 1b*).

**Figure supplement 2.** Quantification of temporal relationship for individual pairs of eyelid Purkinje cells (PCs) and PC-MLIs.

(red points) PCs reached their half-maximal response before putative PC-MLIs did (*Figure 9—figure supplement 1b*). Importantly, this temporal relationship was not observed during trials without CRs (blue points). Next, we aligned firing rate to CR onset and examined this relationship for the entire ensemble of 136 pairs of simultaneous recordings of the activity of putative PC-MLIs and eyelid PCs during conditioning (*Figure 9—figure supplement 2a*). Individual pairs of PCs and putative PC-MLIs were examined by calculating a ratio representing the magnitude of changes in activity prior to CR onset, normalized to the peak amplitude of the change during the entire interval (*Figure 9—figure supplement 2b*). The distribution of these differences, color-coded by ISI, is shown in *Figure 10*. This

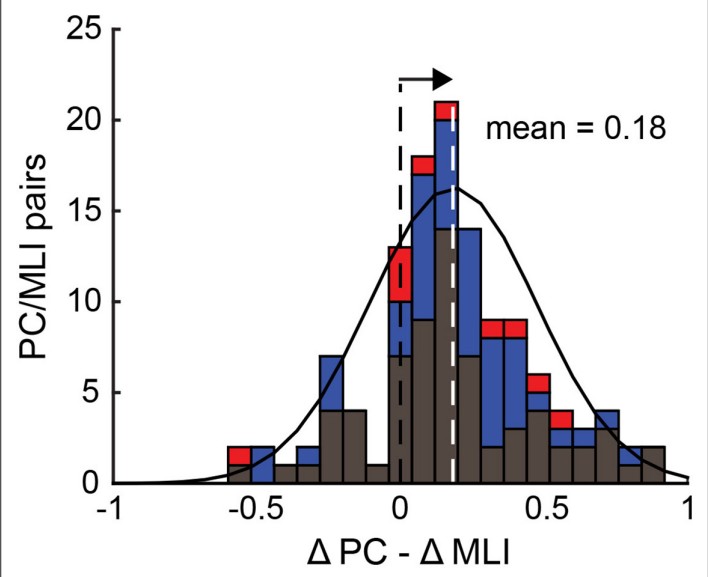

**Figure 10.** Relationship between the responses of simultaneously recorded eyelid PC-MLIs and eyelid PCs and conditioned eyelid responses. Distribution showing relationship between timing of decreases in eyelid PCs activity and increases in PC-MLIs activity at each interstimulus interval (ISI) (black = ISI 250, 76 PC/PC-MLI pairs; blue = ISI 500, 50 pairs; red = ISI 700 and 750, 10 pairs). The mean of the distribution of ΔPC − ΔMLI differences is indicated with a white dashed line. This distribution is shifted (arrow) to the right of 0 (black dashed line), indicating that the PC activity decreases more rapidly than the PC-MLI activity increases, relative to CR onset, for most pairs of neurons. MLIs, molecular layer interneurons; PC, Purkinje cell.

The online version of this article includes the following source data and figure supplement(s) for figure 10:

**Source data 1.** Source files for ΔPC-ΔMLI distribution between the responses of simultaneously recorded eyelid PC and eyelid PC-MLI during eyelid conditioning at different interstimulus intervals (ISIs).

**Figure supplement 1.** Relationship between activity of Purkinje cell (PC) and PC-MLI pairs from simulations featuring different sites of circuit plasticity.

distribution has a mean (white dashed line) that is greater than zero (black dashed line), indicating that eyelid PCs decreased their activity before putative PC-MLIs increased their activity in a majority of cases. A t-test (verified with Jarque-Bera, $p=0.5$) indicated that the combined distribution was significantly different ($p=8 \times 10^{-15}$) from a normal distribution with a mean of 0. Individual ISI distributions were also skewed rightward (skewness = 0.27 for ISI 250, 0.36 for ISI 500, and 0.78 for ISI 750), indicating that, on average, eyelid PC activity decreased before putative PC-MLI activity increased.

In summary, the activity of PCs and putative PC-MLIs is anti-correlated. This is true both during CRs and at other times, and is particularly evident during large changes in PC activity. Further, several different types of analysis all point to the conclusion that changes in PC activity precede both CR onset and changes in putative PC-MLI activity. All these results are consistent with prediction (3) of the simulation. Collectively, our measurements of the activity of PCs and putative PC-MLIs – and their relationship to CRs – are in line with all three predictions of the simulation. Given that the computational predictions require the presence of PC-to-PC-MLI feedback inhibition, we conclude that this feedback circuit likely operates in the cerebellum during motor learning.

### Further insights from circuit simulations

Given the success of the computational simulation in predicting the in vivo properties of PC-MLIs, we further interrogated the simulation to understand the role of feedback inhibition in cerebellar circuit performance during conditioning.

First, we determined the origins of the temporal relationship between changes in eyelid PC activity and PC-MLI activity during conditioning. Because both PC and PC-MLI activity changes during eyelid conditioning, plasticity likely occurs at parallel fiber (PF) synapses somewhere within the circuit; these synapses are known to be plastic and potentially involved in associative learning (**Ito, 1986**;

*Lisberger, 2021*). To deduce where this plasticity occurs, we tested simulations with different loci of learning-induced PF plasticity. Comparisons between the predicted and observed activity of PC-MLIs and PCs (*Figure 10—figure supplement 1*) indicated that the presence of plasticity at PF synapses onto PCs yielded changes in PC activity that preceded changes in PC-MLI activity, as observed in vivo (*Figure 10—figure supplement 1a and b*). However, if plasticity was present only at PF synapses onto MLIs, changes in firing rates of MLIs and PCs were approximately simultaneous, with changes in MLI activity slightly preceding changes in PC activity (*Figure 10—figure supplement 1c and d*). Including plasticity at both synapses produced too many MLIs that led PCs (data not shown), in clear contradiction to the in vivo data. In summary, only simulations with plasticity exclusively at PF excitatory synaptic inputs to PCs were capable of accurately predicting our observed results. Thus, it appears that learning-related plasticity may occur at the PF-to-PC synapse, rather than PF-to-MLI synapses.

Next, we considered the behavioral consequences of the feedback circuit from PCs to PC-MLIs via additional simulations that predicted the ability of the cerebellum to learn during eyelid conditioning. In these simulations, eyelid conditioning was mimicked by presenting mossy fiber and climbing fiber inputs over a wide range of ISIs. Examples of virtual eyelid responses, produced by simulations trained at an ISI of 200 ms, are shown in *Figure 11a*. It can be observed that while simulations with or without PC feedback learned, the simulation that included PC feedback performed better (*Figure 11a*, right). This was true at nearly all ISIs: for example, in *Figure 11b and c* only simulations with PC feedback were able to acquire CRs at the shortest (150 and 200 ms) and longest ISIs (ISI >700 ms). Even at inter-mediate ISIs, where simulations were able to learn without PC feedback, the presence of PC feedback improved the rate of acquisition and final asymptotic performance. This is evident from comparison of CR amplitude acquisition curves at ISI 500 ms in simulations with (black) and without (red) PC feedback (*Figure 11b*). These results indicate that PC feedback can increase the rate of learning, particularly improving performance on the temporal margins, where stimulus patterns otherwise make learning difficult.

Finally, to better understand how PC feedback improves learning, from the simulation results we calculated training-induced changes at all 12,000 excitatory PF synapses between granule cells and PCs. In *Figure 12a–d*, synapses that underwent net LTD appear as negative changes in synaptic weight, while those showing LTP have positive changes in weight. While most synapses underwent little or no net change in strength, more PF synapses underwent LTD than LTP in simulations both with and without PC feedback (*Figure 12a and b*). However, comparing the outcome with and without feedback (*Figure 12c*) revealed that fewer PF synapses underwent LTD in the presence of PC feedback. Subtraction of the results from the two simulations reveals the net decrease in learning-dependent LTD of PF synapses resulting from PC feedback (*Figure 12d*). To compare this effect across all 11 ISIs, we converted different histograms, such as the one in *Figure 12d*, into cumulative distributions of changes in synaptic weights (*Figure 12e*); the blue line (ISI 500) is derived from the data shown in *Figure 12d*. For all ISIs, PC feedback is predicted to cause fewer synapses to undergo LTD (*Figure 12e*). In summary, simulations with PC feedback learned better and required less net LTD of PF synaptic efficacy, indicating a more efficient distribution of learning-induced changes in synaptic strength.

## Discussion

We used optogenetic circuit mapping and paired electrophysiological recordings in cerebellar slices to demonstrate functional inhibitory connections between PCs and PC-MLIs and to characterize the physiological properties and spatial organization of this novel synaptic circuit. Our optogenetic circuit mapping enabled the first detailed analysis of the function and spatial organization of this circuit. A large-scale simulation of the cerebellum predicted that this interconnectivity should cause PC-MLIs to exhibit several unique properties during eyelid conditioning, a cerebellum-mediated form of motor learning. In vivo recordings from MLIs during delay eyelid conditioning, a well-established cerebellum-mediated task, tested and validated the predictions of the simulation. Finally, additional simulations suggested that inhibitory feedback from PCs to PC-MLIs increases cerebellar efficiency by permitting learning with less overall net synaptic plasticity.

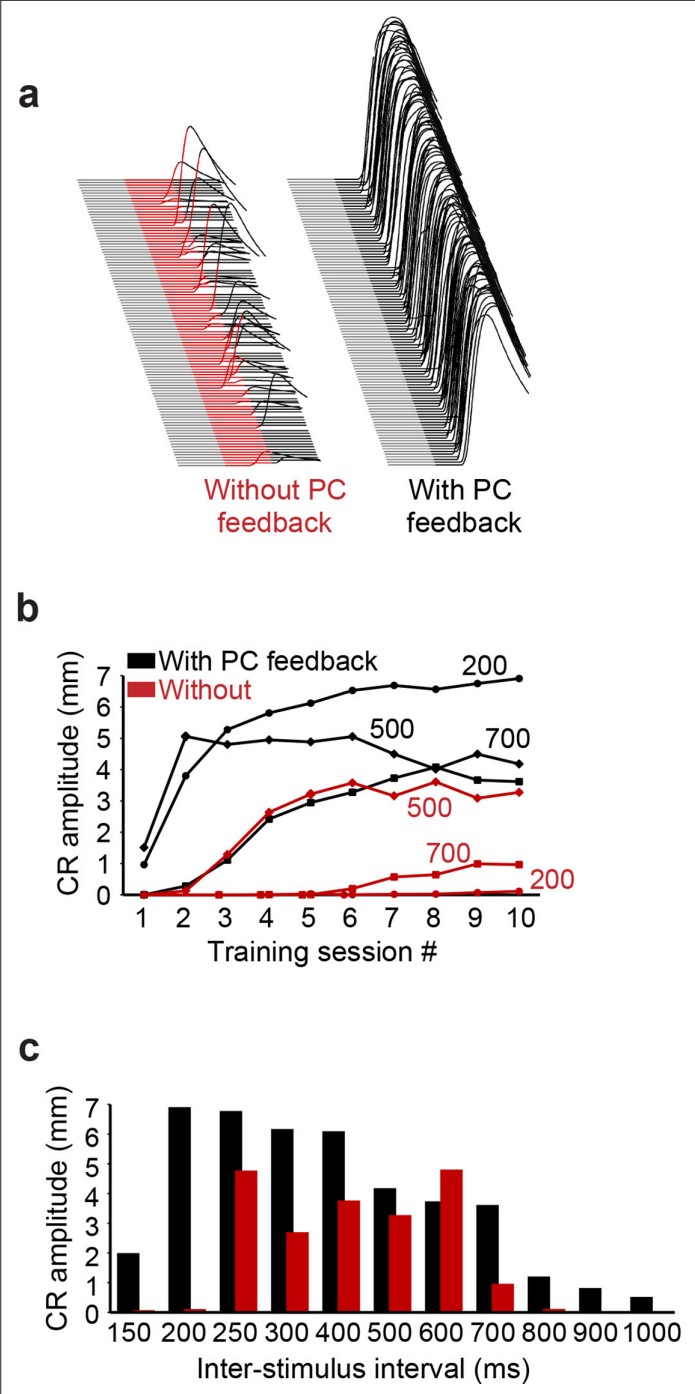

**Figure 11.** Simulation performance with and without Purkinje cell (PC) feedback to molecular layer interneurons (MLIs). (**a**) Examples of virtual conditioned responses (CRs) from the last session of training at interstimulus interval (ISI) 200 for simulations without feedback from PCs to MLIs (conditioned stimulus [CS] is shown in red) and with feedback collaterals (CS is shown in black). (**b**) Acquisition curves for simulations constructed with (black) and without (red) feedback from PCs to PC-MLIs, plotting the amplitude of virtual CRs as a function of sessions of training for the three ISIs indicated (108 trials per session). (**c**) Average virtual CR amplitude on the final session of training is shown for training using 11 ISIs ranging from 150 to 1000 ms; black bars indicate results from simulations with PC to PC-MLI feedback and red bars depict data from simulations lacking feedback. Simulations with PC feedback performed considerably better at relatively long and short ISIs, while for intermediate ISIs there was less effect on performance.

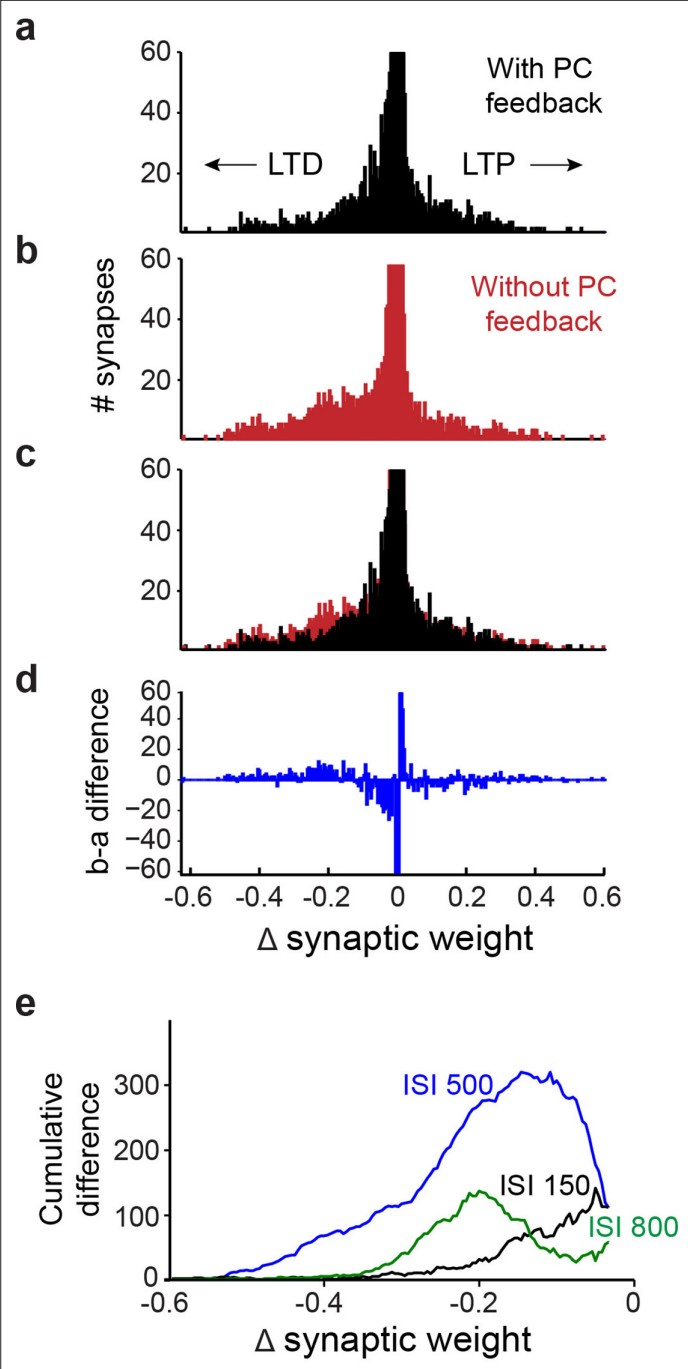

**Figure 12.** Simulation of learning-dependent plasticity with and without Purkinje cell (PC) feedback. (**a**) Net changes in the strength of the 12,000 synapses between granule cells and PCs in a simulation with PC feedback to PC-MLIs and trained at an interstimulus interval (ISI) of 500ms. Negative changes indicate net LTD (left), while positive changes indicate net LTP (right). The vertical scale of the histogram is cropped at 60 synapses because the vast majority of synapses showed little or no change. The synapses that underwent net LTD are most likely to support expression of the conditioned responses in the simulations. (**b**) Same analysis as in **a**, for a simulation lacking PC feedback to PC-MLIs. (**c**) Superimposition of **a** and **b** indicates a larger number of synapses underwent robust LTD in the absence of PC feedback. (**d**) Difference between **a** (with feedback) and **b** (without feedback). The simulation lacking PC feedback to PC-MLIs had a larger number of synapses undergoing robust LTD and fewer synapses undergoing a small degree of net LTD. (**e**) To compare the differences between simulations with and without PC feedback across several ISIs, difference plots such as the one shown in **d** were converted to a cumulative histogram. Across the three ISIs shown, the simulations lacking PC feedback collaterals had more

*Figure 12 continued on next page*

*Figure 12 continued*

synapses undergoing robust LTD than those with feedback. This was true for ISIs where the simulations learned approximately equally well (ISI 500 in blue) and for ISIs where simulations with collaterals learned much better (ISIs 150 and 800). MLIs, molecular layer interneurons.

## Functional organization of PC-MLI circuits

It has long been known that PC collateral axons bifurcate and extend along the PC layer (*Hámori and Szentágothai, 1968*; *Larramendi and Lemkey-Johnston, 1970*; *Chan-Palay, 1971*; *O'Donoghue and Bishop, 1990*; *Witter et al., 2016*), to inhibit other PCs and to reach into the lower molecular layer and upper granule cell layer. In rodents, PC axon targets outside of the PC layer are mainly MLIs within the molecular layer as well as Lugaro cells and granule cells in specialized regions of the granule cell layer (*Larramendi and Lemkey-Johnston, 1970*; *Chan-Palay, 1971*; *Guo et al., 2016*; *Witter et al., 2016*). Here, we built upon this anatomical information, and limited physiological analysis done in vivo (*Bloedel and Roberts, 1969*; *O'Donoghue et al., 1989*) or in slices (*Witter et al., 2016*), to functionally characterize the monosynaptic connection between PCs and MLIs.

Connections between PCs and MLIs are selective: PCs apparently connect to only a fraction of MLIs within the inner one-third of the molecular layer. It remains to be determined whether the MLIs that receive PC feedback inhibition are basket cells or some other type of interneuron. Our connectivity rate is lower than a previous observation (*Witter et al., 2016*); this difference may arise from our more reliable (seven times larger) sample size. Although the fraction of MLIs receiving PC input could be underestimated, due to loss of connections during slicing of the cerebellum, our observations are consistent with anatomical observations of sparse representation of PC axons within the molecular layer (*Chan-Palay, 1971*; *Watt et al., 2009*). Our observation of no functional connectivity between PCs and stellate cells in the outer two-thirds of the molecular layer is also consistent with anatomical studies (*Hámori and Szentágothai, 1968*; *Lemkey-Johnston and Larramendi, 1968*). These connectivity features account for the difficulty in finding connected pairs of neurons, both in vitro and in vivo.

We were able to characterize the fundamental properties of the PC-to-PC-MLI circuit. Inhibitory synaptic responses evoked by PC stimulation were sufficiently strong to prevent action potential firing in postsynaptic PC-MLIs. These responses had a short onset latency and were unaffected by a glutamate receptor antagonist, consistent with a monosynaptic connection. Other noteworthy properties of IPSCs at this synapse include variable amplitudes and fast rise times. These properties generally are similar to those of IPSCs at the inhibitory synapse formed between PCs via axon collaterals (*Orduz and Llano, 2007*; *Watt et al., 2009*; *Witter et al., 2016*), aside from a lower rate of failures at the PC-to-PC-MLI synapse. The latter property arises from a relatively large RRP of synaptic vesicles. A remarkable activity-dependent enhancement of mobilization of reserve pool synaptic vesicles allows this synapse to sustain transmission even at the high rates of activity typical of PCs.

Our high-speed mapping visualized the spatial organization of PC inputs onto a postsynaptic PC-MLI and revealed one or two discrete areas of input within the molecular and PC layers, where PCs are located. The presence of PC-MLI receiving two discrete areas of presynaptic PC input (e.g. *Figure 3d2*), as well as a mean presynaptic input area 1.5 times larger than the size of the optical footprint of a single PC (*Figure 3e*), indicates that on average one or two PCs converge onto a postsynaptic MLI. This is in line with the relative abundance of PCs and PC-MLIs: our measurements indicate that approximately 1/18 of all MLIs receive PC feedback, while previous estimates suggest that the number of PCs is approximately 1/10 that of MLIs. Thus, PCs are 1.8 times more abundant than PC-MLIs, which is in reasonable agreement with the 1.5-fold convergence between PCs and PC-MLI that we observed. Because PCs are the sole output of the cerebellar cortex, any cell that influences PCs will influence behavior; the comparable numbers of PCs and PC-MLIs indicates high potential for such influences. The relative abundance of PCs and PC-MLI also suggests that nearly every PC provides feedback inhibitory drive to PC-MLIs. Our observation that PC-MLIs sometimes receive two convergent PC inputs (*Figure 3d2*) contrasts with a previous suggestion that only single PCs innervate MLIs (*O'Donoghue et al., 1989*). PC-MLI connections were mostly located within 200 μm from postsynaptic PC-MLI somata, consistent with the reported sagittal extent of PC axon collaterals (*O'Donoghue and Bishop, 1990*; *Watt et al., 2009*; *Witter et al., 2016*). We also found that a majority of presynaptic PCs were located on either the apical or juxtaposed side relative to postsynaptic MLIs, verifying anatomical observations (*Watt et al., 2009*; *Witter et al., 2016*).

Despite using two experimental strategies that optimized detection of either PC-to-MLI or MLI-to-PC connectivity, in no case were reciprocally connected pairs of PCs and MLIs observed (*Figure 4c*). Although it was previously assumed that PC-to-PC-MLI circuits are non-reciprocal (*O'Donoghue et al., 1989*; *Guo et al., 2016*), our results provide the initial experimental test of this important point. Such an arrangement allows feedback inhibition of PC-MLIs by PCs to disinhibit neighboring PCs (*Figure 4d*), which may help to synchronize PC activity in the sagittal plane. It will also serve to prevent network oscillations, which are often the outcome of reciprocal inhibitory circuits (*Satterlie, 1985*).

## In vivo activity of MLIs

This interconnectivity between PCs and MLIs gave rise to specific and testable predictions about the activity of PC-MLIs in vivo. MLIs that receive input from PCs should fire inversely with PCs (*Figure 4e*) and, like PCs, their activity should correlate on a trial-by-trial basis with cerebellar-mediated behavioral responses. We tested these predictions using eyelid conditioning because: (1) the role of the cerebellum in this task is well established; (2) the activity of eyelid PCs is well characterized and appears to control the timing and amplitude of the behavioral CRs; and (3) computer simulations of the cerebellum allowed us to explore the plausibility of alternative explanations for why MLI activity correlates, on a trial-by-trial basis, with PC activity and behavioral responses.

Recordings from MLIs during conditioned eyelid responses supported these predictions. A small fraction of MLIs showed unusually high trial-by-trial correlations with the eyelid responses. Computer simulations predicted such high correlations only when PC-to-PC-MLI connections were included, even when these simulations were constructed to exaggerate contributions from other sources. Simultaneous recordings showed that changes in eyelid PC activity tended to precede, on a trial-by-trial basis, that of putative PC-MLIs (*Figures 9a–c and 10*). Cross-correlation analysis revealed similar inverse activity correlations between PCs and PC-MLIs across all three platforms we examined: cerebellar slices (*Figure 4e*), in vivo recordings (*Figure 8a*), and computational modeling (*Figure 8b*). Finally, we observed that computer simulations trained with eyelid conditioning procedures learned better when PC-to-MLI connections were included (*Figure 11*), despite these connections yielding less net synaptic plasticity (*Figure 12*). Even though previous versions of our simulation that did not include PC-to-MLI feedback could exhibit learning (*Medina and Mauk, 2000*; *Li et al., 2013*), simulations that included PC-MLI connections performed better, particularly in difficult learning conditions.

These results suggest that in networks where PCs inhibit MLIs, each PC acts to make other PCs fire more like itself through disinhibition (*Figure 4d*). In this way, PC-MLIs can exert strong influence over cerebellar processing and learning despite being a small fraction of MLIs. Because PC-MLIs are both inhibited by PCs and inhibit other PCs, their connectivity confers upon them particularly strong influence over PC activity and thence over the output of the cerebellar cortex. We observed inverse activity between PCs and PC-MLIs during instances of large increases or decreases in PC activity, both during large pauses in PC activity controlling expression of CRs and during the largest changes in PC activity during the inter-trial interval (pause/burst-triggered averages). As a consequence, PC feedback to PC-MLIs could serve to selectively communicate large changes in PC activity within a parasagittal strip through PC-MLIs.

Inhibitory neurons are ubiquitous throughout the brain and are often connected to each other by inhibitory synapses. Therefore, our results suggest a computational principle: activation that involves disinhibition – as occurs between PCs and PC-MLIs – could synchronize the activity of inhibitory neurons. Here, we use 'synchrony' as a broad term that includes PC pause synchronization. Pause synchrony can be exerted as diverse forms (pause beginning, pause ending, pause overlapping synchrony) for several milliseconds, which results in a variety of forms of modulation of neuronal activity and may lead to a broad range of behavioral outcomes (*Sudhakar et al., 2015*; *Hong et al., 2016*). A transient increase in PC synchrony recently reported to occur during saccades (*Sedaghat-Nejad et al., 2022*) also occurs over a timescale that is compatible with our observations of PC feedback and feedforward inhibition.

One apparent consequence of disinhibition-based synchronization of learning-related activity in PCs suggested by our simulations is to require a smaller net change in excitatory synaptic input for PCs to decrease their activity to the level required to produce a well-timed CR in response to the CS. Such economy of plasticity could be greatly beneficial for faster learning, better learning at difficult ISIs, and increased capacity to store learned responses. While it is well established that plasticity occurs

at PF synapses between GCs and PCs (*Ito, 1986*; *Lisberger, 2021*) (blue in *Figure 5a*), such plasticity cannot be directly proven based on measurements of learning-related changes in PC activity. For example, these changes could instead result from plasticity at synapses onto MLIs (green in *Figure 5a*). However, such plasticity has only been inferred from learning-related changes in MLI activity and from MLI receptive field expansion (*Jörntell and Ekerot, 2002*; *Jörntell et al., 2010*). Indeed, previous evidence of learning-dependent changes in MLI activity during eyelid conditioning was also interpreted to result from plasticity at MLI synapses (*ten Brinke et al., 2015*). Our results substantially limit such inferences by showing that learning-dependent changes in MLI activity could instead come from interactions between PCs and PC-MLIs. While our results do not completely exclude the possibility of learning-related changes at PF synapses innervating MLIs, they do demonstrate that learning-related changes in the activity of MLIs (or PCs) do not necessarily indicate changes in the strength of synapses onto MLIs (or PCs). Further work will be required to fully sort out the contributions of MLI synaptic plasticity to learning. Independent of detailed mechanism, we nonetheless conclude that feedback inhibition from PCs to PC-MLIs contributes to more efficient cerebellar-mediated motor learning.

## Methods

**Key resources table**

| Reagent type (species) or resource | Designation | Source or reference | Identifiers | Additional information |
|---|---|---|---|---|
| Genetic reagent (*Mus musculus*) | Pcp2-cre mice; B6.129-Tg(Pcp2-cre)2Mppin/J | Jackson Laboratories | Stock No: 004146; RRID:IMSR_JAX:004146 | |
| Genetic reagent (*Mus musculus*) | Pcp2-cre mice; B6.Cg-Tg(Pcp2-re)3555Jdhu/J | Jackson Laboratories | Stock No: 010536 RRID:IMSR_JAX:010536 | |
| Genetic reagent (*Mus musculus*) | Ai32 mice; B6;129S-*Gt(ROSA)26Sor*$^{tm32(CAG-COP4*H134R/EYFP)Hze}$/J | Jackson Laboratories | Stock No: 012569 RRID:IMSR_JAX:012569 | |
| Chemical compound, drug | Bicuculline | Sigma-Aldrich | Cat#14340 | |
| Chemical compound, drug | Gabazine | Sigma-Aldrich | Cat#S106 | |
| Chemical compound, drug | Kynurenic acid | Sigma-Aldrich | Cat#K3375 | |
| Chemical compound, drug | QX-314 (Lidocaine *N*-ethyl bromide) | Sigma-Aldrich | Cat#L5783 | |
| Chemical compound, drug | Alexa 594 | Thermo Fisher | Cat#A10438 | |
| Software, algorithm | pClamp 10 | Axon instruments | RRID:SCR_011323 | |
| Software, algorithm | Fiji | NIH | RRID:SCR_002285 | |
| Software, algorithm | Origin | Origin Lab | RRID:SCR_014212 | |
| Strain, strain background (*Oryctolagus cuniculus*, males) | New Zealand albino rabbits | Myrtle's Rabbitry | | |
| Software, algorithm | Cheetah Data Acquisition | Neuralynx | Cheetah 5.0 | |
| Software, algorithm | Spike-Sorting Program | M.D.M. | WinClust | |
| Software, algorithm | Experimental Control | M.D.M. | BunTrain | |
| Other | .0005 NiCr wire | Kanthal | M#756027 | Tetrode wire |
| Other | 16 tetrode EIB | Neuralynx | EIB-36-16TT | Electrode interface board |

## In vitro electrophysiological recordings and optogenetic mapping
### Mice

PCP2 (Purkinje cell protein 2 or L7)-ChR2-H134R mice were generated as previously reported. PCP2-cre transgenic mice [(Pcp2-cre)2Mpin/J or (Pcp2-cre)3555Jdhu/J; Jackson Labs] (*Barski et al., 2000*; *Asrican et al., 2013*) were crossed with mice expressing floxed ChR2-H134R [B6;129S-Gt(ROSA)26Sor$^{tm32(CAG-COP4*H134R/EYFP)Hze/J}$] (*Madisen et al., 2012*). Mice positive for both transgenes were

selected through PCR-based genotyping (*Wang et al., 2007*) using the following primers: forward primer; 5'-GCG GTC TGG CAG TAA AAA CTA TC-3' and reverse primer; 5'-GTG AAA CAG CAT TGC TGT CAC TT-3' (*Asrican et al., 2013*). Mice were maintained under a 12 hr light/dark cycle with free access to food and water. All procedures were conducted according to the Institutional Animal Care and Use Committee guidelines of the Biopolis Biological Resource Center.

## Brain slice recording

Conventional methods were used to prepare cerebellar slices from mice aged between 14 and 77 days (*Wang et al., 2007*; *Kim et al., 2014*). Mice were sacrificed by decapitation under deep halothane anesthesia and the brain was rapidly removed. Sagittal cerebellar slices (300 µm thickness) were cut with a Leica vibratome in ice-cold oxygenated cutting solution containing (in mM): 250 sucrose, 26 $NaHCO_3$, 10 glucose, 4 $MgCl_2$, 3 myo-inositol, 2.5 KCl, 2 sodium pyruvate, 1.25 $NaHPO_4$, 0.5 ascorbic acid, 0.1 $CaCl_2$, 1 kynurenic acid. After cutting, slices were kept for at least 1 hr in the oxygenated standard external solution which contained (in mM): 126 NaCl, 24 $NaHCO_3$, 1 $NaH_2PO_4$, 2.5 KCl, 2.5 $CaCl_2$, 2 $MgCl_2$, 10 glucose, 0.4 ascorbic acid (pH 7.4 when equilibrated with 95% $O_2$/5% $CO_2$). In some experiments, bicuculline (10 µM; Sigma), gabazine (SR-95531, 5 µM; Sigma), or kynurenic acid (2 mM; Sigma) was added to the external solution to block chemical synapses.

Whole-cell patch clamp recordings were performed at room temperature, unless otherwise indicated, while the slices were in a recording chamber continuously perfused with extracellular solution. Patch pipettes for PC (3–6 MΩ) and MLI (6–10 MΩ) recordings were pulled on a PC-10 puller (Narishige, Japan). Pipettes were filled with a solution containing (in mM): 130 K-gluconate, 10 KOH, 2.5 $MgCl_2$, 10 HEPES, 5 EGTA, 4 $Na_2ATP$, 0.4 $Na_3GTP$, 5 disodium phosphocreatine (pH 7.3, ~295 mOsm). Membrane potentials were not corrected for liquid junction potentials. Alexa 594 (50 µM) was added to the internal solution to image cell morphology. In a subset of interneuron recordings, QX-314 (10 mM; Sigma) was included in the internal solution to block MLI action potentials. Because the IPSC reversal potential was –74 mV (*n*=3), IPSCs were recorded at a holding potential of –40 mV unless otherwise indicated. Electrical signals were acquired via a Multiclamp 700B amplifier (Molecular Devices) and digitized at 100 kHz via a Digidata 1440A interface (Molecular Devices).

To measure synaptic depression, three different stimulation methods were used: paired electrophysiological recordings, optogenetic stimulation, and extracellular stimulation. Because photostimulation could only evoke PC activity up to 25 Hz (*Wang et al., 2007*) and paired PC-PC-MLI recordings were difficult to obtain, we primarily used extracellular stimulation. To do this, a glass pipette filled with external solution was placed in the granule cell layer below the location of a presynaptic PC, identified by optogenetic mapping. These experiments were done at 34–35°C, to more closely emulate physiological conditions, and kynurenic acid (2 mM) was used to block glutamatergic inputs to PC-MLIs.

## Photostimulation

PCs were photostimulated as previously described (*Kim et al., 2014*). For wide-field excitation, blue light (465–495 nm) illuminating a relatively large area (~0.233 mm[2]) of brain slices was provided by a mercury arc lamp; light pulse duration was controlled by an electronic shutter (Uniblitz T132; Vincent, Rochester, NY). To perform high-speed circuit mapping, a laser-scanning microscope (FV1000MPE; Olympus, Tokyo, Japan) equipped with 25×, NA 1.05 (Olympus XLPlan N) water-immersion objective lens was used to create small spots of laser light to photostimulate ChR2-expressing PCs. A 510×510 µm[2] area of the slice was scanned with a 405 nm laser spot (4 ms duration) in a 32×32 array of pixels, yielding a scanning resolution of 16 µm. The laser spot was scanned in a pseudorandom sequence, to avoid photostimulation of adjacent pixels, while cellular responses were simultaneously measured in whole-cell patch clamp recordings. The same microscope was used for confocal imaging of neurons filled with Alexa 594.

In the course of our experiments, we used approximately 140 Pcp2-Cre Mpin mice (*Barski et al., 2000*). In four of these mice (3%), we saw expression of ChR2 in MLI (and in other cells beyond PCs), readily evident as measurable photocurrents in response to light stimulation, consistent with a previous observation (*Witter et al., 2016*). These mice were discarded and were not included in our analyses. Even in these four mice, the laser intensity required to evoke action potentials in MLIs

non-specifically expressing ChR2 was greater than 6 µW; because we used a laser intensity of 3 µW in our mapping experiments, there would have been no possibility of evoking action potentials in MLI. Among the remaining 136 mice that we did use for our experiments, we recorded from more than 400 MLI and never detected photocurrents. Thus, there was no ChR2 expression in MLIs under our experimental conditions. We also obtained similar results in another PCP2-cre Jdhu line (*Zhang et al., 2004*), further indicating that our optical circuit mapping precisely detected the connectivity between PCs and PC-MLIs.

## Data analysis

Optogenetic mapping data were analyzed with custom software written in MATLAB by P Namburi as previously described (*Kim et al., 2014*). Spatial maps of the light sensitivity of ChR2-expressing PCs (optical footprints) were created by correlating the location of the photostimulation spot with action potentials evoked between 0 and 14 ms after the start of a light pulse (4 ms duration). Due to the relatively high rate of spontaneous activity of PCs, typically two or three maps were averaged to reduce the influence of this background activity: only locations where action potentials were consistently evoked were included within optical footprints.

To generate synaptic input maps, the location of the light spot was correlated with the resulting IPSCs in MLIs. The minimum threshold amplitude for detecting IPSCs was 15 pA. The relatively high rate of spontaneous IPSCs created background noise; this was reduced by averaging together two to three input maps, with pixels having an IPSC detection probability of 0.5 or greater considered to arise from photostimulation of presynaptic PCs, rather than spontaneous activity. The area of the input field was measured by summing all pixels showing responses meeting these criteria. Single-pixel areas within input maps also result from spontaneous PC activity and also were excluded from analysis (*Kim et al., 2014*). The peak amplitude and latency of IPSCs were determined from the maximum IPSC measured between 4 and 24 ms after the start of the light pulse. Onset times were estimated as the zero-crossing time of a line joining values measured from 20% to 80% of the IPSC peak. Throughout, data are expressed as mean ± SEM.

## In vivo recordings and behavioral training methods
### Rabbits

Ten male New Zealand albino rabbits (Oryctolagus cuniculus; Myrtle's Rabbitry), weighing 2.5–3 kg at experiment onset, were used for in vivo experimentation. Treatment of rabbits and surgical procedures were in accordance with National Institutes of Health guidelines and an institutionally approved animal welfare protocol. All rabbits were maintained on a 12 hr light/dark cycle.

### Surgery

One week before the start of recording, rabbits were removed from their home cage and anesthetized with a cocktail of acepromazine (1.5 mg/kg) and ketamine (45 mg/kg). After onset of anesthesia, animals were placed in a stereotaxic frame and maintained on isoflurane (1–2% mixed in oxygen) for the remainder of the surgery. Under sterile conditions the skull was exposed with a midline incision (~5 cm), and four holes were drilled for screws that anchored the head bolt in place. The animal's head was then positioned with lambda 1.5 mm ventral to bregma and a craniotomy was drilled out at 5.9 mm posterior and 6.0 mm lateral to lambda. The skull surface was marked and skull fragments were carefully removed from the craniotomy along with the dura matter under visual guidance. A custom-made hyperdrive array (12 or 16 tetrodes, 2 references) fitted with an electronic interface board (EIB-54 or EIB36-16TT, Neuralynx) was implanted in the left anterior lobe of the cerebellar cortex at 17.8 mm ventral to lambda. Final ventral placement of tetrodes during surgery was between 1 and 2 mm above the target coordinate to allow advancement of the tetrodes to the target. Hyperdrives were positioned at a 40° angle caudal to vertical to avoid the cerebellar tentorium. This region of the cerebellum has been shown to be involved in acquisition and expression of well-timed conditioned eyelid responses in rabbits (*Kalmbach et al., 2011*). The bundle cannula of the hyperdrive was

lowered to the surface of the brain and the craniotomy was sealed with low viscosity silicon (Kwik-Sil; World Precision Instruments). A screw attached to an insulated silver grounding wire (0.003" bare, 0.0055" coated, A-M Systems) was then screwed into the skull. The silver wire was also attached to the ground channel of the EIB with a gold pin. The head bolt, screws, and hyperdrive were secured with dental acrylic (Fastray Pink; The Harry J Bosworth Company), and the skin was sutured where the skull and muscle was exposed. Finally, two stainless steel stimulating electrodes were implanted subdermally caudal and rostral to the left eye. Rabbits were given analgesics and antibiotics for 2 days after surgery and monitored until fully recovered.

## Tetrode recording and unit isolation

Each independently movable tetrode was composed of four nichrome wires (12 µm diameter; Kanthal Palm Coast), twisted and partially melted together to form a tetrode. Individual wires of each tetrode were connected to the electronic interface board (EIB) with gold pins, all four wires of the reference tetrodes were connected to a single reference channel of the EIB. Each tetrode was gold plated to reduce its final impedance to 0.5-1.5 MΩ measured at 1 kHz (impedance tester IMP-1; Bak Electronics). Tetrodes targeting cerebellar cortex were placed over the left anterior lobe and advanced to within 2.0 mm of the target during surgery using stereotaxic guidance. Tetrodes were then lowered in 40–80 µm increments during turning sessions (~1 hr) until at least one stable single unit was identified; there were often multiple units on a single tetrode. After turning sessions, tetrodes were allowed to stabilize for at least 24 hr and units were checked again before recording and behavioral training commenced. Putative PCs were initially identified by their higher baseline firing rate relative to cerebellar cortical interneurons and later confirmed by identifying complex spikes during cluster cutting. PCs with complex spikes evoked by eyelid stimulation were operationally defined as eyelid PCs (*Halverson et al., 2015*). MLI single units were often recorded simultaneously on the same or different tetrodes during recordings from PCs. Recordings were done once a single unit was identified and stable without knowledge of the origin of its climbing fiber input for PCs or identity of MLI type. Tetrodes in cerebellar cortex targeted the region containing PCs receiving evoked complex spikes from eyelid stimulation (*Figure 13a*). The use of tetrodes allowed for simultaneous recording of multiple single units, including MLIs, during conditioning experiments. All PCs included in the analysis had confirmed spontaneous complex spikes and all single units were held throughout the entire session. Any recording that was lost during a recording session was not included in the analysis.

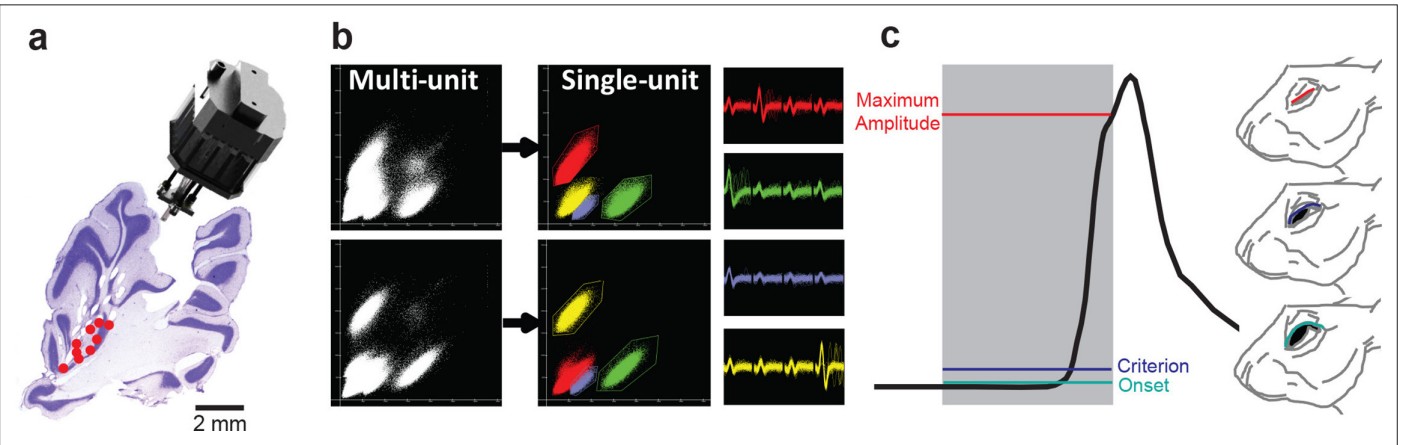

**Figure 13.** Single-unit recordings in cerebellum and eyelid conditioning in the rabbit. (**a**) Sagittal section of the cerebellum showing tetrode tracks and the final location (red points) of tetrodes that recorded eyelid molecular layer interneurons (MLIs) and eyelid Purkinje cells (PCs). The custom-made hyperdrive array is shown above the cerebellar section. (**b**) Example of single units being isolated from a multi-unit recording with cluster cutting. The individual clusters and waveforms are color-coded to illustrate how unique features of the waveform across the four channels of the tetrode can be used to isolate single units from the multi-unit recording. (**c**) An example of a single conditioned eyelid response trial where the different response measures are color-coded according to the position of the eyelid in the rabbit. Gray shading indicates the duration of the conditioned stimulus and the black line represents the position of the eyelid throughout the trial. The conditioned response is delayed relative to stimulus onset and peaks at the end of the interstimulus interval (ISI). Different response measurements are color-coded according to eyelid position (right).

Neuronal signals were first preamplified at unity gain and then fit to a window between 250 and 2000 µV and bandpass filtered (0.3–6 kHz; Neuralynx). Signals that exceeded a channel amplitude threshold were digitized and stored at 32 kHz (Cheetah system; Neuralynx). Custom interactive cluster cutting programs were used to manually isolate and identify single units (*Figure 13b*). Waveform characteristics were plotted as a two-dimensional scatter plot of one wire of the tetrode versus another in terms of energy, peak, and valley measures. The energy measure represents the square root of the sum of the squared points for the entire waveform. The peak measure is the maximum height (positive amplitude) of the waveform. The valley measure is the maximum depth (negative amplitude) of the waveform. When possible, initial identification of a single unit was made using the peak as channel thresholds were set during recordings with that feature of the waveform. For recordings in cerebellar cortex a late peak measure, defined as the maximum peak during the last five points of the 32 points that make up each waveform, was also used to identify the later peak component of the complex spikes from the earlier peak of the simple spikes. From these cluster cutting analyses, a single PC recording would then yield two clusters, one containing simple spikes cut using peak, valley, and energy, and the second containing the complex spikes cut using the late peak parameter (*Halverson et al., 2015*). Following cluster cutting, all subsequent data analysis was performed using custom-written scripts in MATLAB.

## Conditioning procedure

Conditioning experiments and recordings were done in custom training chambers (89 × 64 × 49 cm³). Rabbits were placed in a plastic restrainer and the ears were stretched over a foam pad and taped down to limit head movement. An adjustable infrared emitter/detector was secured in place with the head bolt and aligned to the middle of the left eye. The infrared emitter/detector measured eyelid position by converting to a voltage signal the amount of emitted infrared light reflected back to the detector, which increased as the eyelid closes. The signal was amplified to yield a signal that was linearly related to upper eyelid position (±0.1 mm). The eyelid position detector was then calibrated before each training session by delivering a test US to elicit maximum eyelid closure (6.0 mm). The corresponding voltage deflection (~6 V) was then divided by 6 mm to obtain a mm/V calibration. Each training chamber was also equipped with a speaker connected to a stereo equalizer and receiver which were connected to a computer that generated the tone CS. The CS used during training was either a 1 or 9.5 kHz sinusoidal tone (85 dB), which ramped at onset and offset with a 5 ms time constant to avoid audible clicks from the speaker. To deliver the US, leads from a stimulus isolator (Model #2100, A-M Systems) were attached to electrodes caudal and rostral to the eye. The US was eyelid stimulation, which consisted of trains of 1 ms current pulses delivered at 100 Hz for 50 ms. The intensity was adjusted for each animal to be just above threshold to elicit a full eyelid closure (between 0.8 and 1.5 mA, depending on the condition of the implanted wires).

Stimulus presentation was controlled by custom software operated on a Windows XP-based computer. To permit temporal alignment of neural and behavioral responses, digital timing pulses were generated by the computer controlling stimuli and measuring behavior and were sent to the digital input port on the Digital Lynx acquisition system (Neuralynx). During initial paired delay conditioning the tone CS was 550 ms which co-terminated with the 50 ms eyelid stimulation US which produced an ISI of 500 ms. All rabbits were initially trained and extinguished with delay conditioning at ISI 500. Further conditioning sessions involved either a 1 or 9.5 kHz tone at ISIs of 250, 700, or 750 ending with the same 50 ms eyelid stimulation US. Each training session consisted of 12 nine-trial blocks (108 trials) with each block starting with a CS alone trial followed by eight paired CS-US trials. The mean intertrial interval was 30 s with a range of 20–40 s.

## Eyelid position analysis

For each trial, 2500 ms of eyelid position (200 ms pre-CS, 2300 ms post CS) were collected at 1 kHz and at 12 bit resolution. Data were stored to a computer disk for subsequent off-line analysis. Eyelid position data was passed through a low-pass filter. Response measures calculated for each trial included CR amplitude, latency to CR criterion, and latency to CR onset (*Figure 13c*). CR amplitude was defined as the value of eyelid position from the baseline at the time of US onset. Latency to CR

criterion was defined as the time point at which the CR reached the 0.3 mm criterion to be designated as a CR. Latency to CR onset was determined using a custom-written two-step algorithm. The first step was designed to detect the initial deflection away from the pre-CS baseline, while the second step used linear interpolation to determine the exact time of CR onset. For further analyses eyelid trajectories were truncated at US onset to exclude non-cerebellar influence on the eyelid movement.

## Single-unit recording analysis

Instantaneous firing rate of each single-unit recording was estimated on every trial using a one-sided Gaussian kernel with a 25 ms standard deviation window. We chose a one-sided Gaussian to prevent neural responses related to the US from contaminating unit activity during the CS. PCs firing rate was normalized by the value of the baseline firing rate during 1500 ms of pre-CS activity. Each MLI firing rate was normalized, however, to the maximum during the CS of the average firing rate through the session. Most putative PC-MLI had a similar baseline firing rates, but showed variable CR-related increases in activity, ranging from twofold increases to eight- to tenfold increases over baseline firing rates. Under these conditions, calculating the simple mean of all PC-MLI activity or normalizing it to the baseline firing rate would yield overrepresentation of PC-MLIs with the highest CR-related increases in activity. To avoid such bias, we normalized PC-MLI responses to the maximum mean firing rate during the CS.

For each MLI single unit, cross-correlations between firing rate and behavioral responses were calculated on every trial when the animal produced a CR. For every trial, starting from 150 ms before CS onset and through US onset, we calculated cross-correlations between the instantaneous firing rate profile and the time profile of the CR (eyelid position). Then, cross-correlation values were averaged through CR trials to produce an average single-trial correlation with behavior for a given MLI (see *Figure 5*). For comparison of activity of simultaneously recorded eyelid PCs and MLIs, cross-correlations were calculated between firing rates then averaged through the session (see *Figure 10*). In all cases, mean values of arguments were subtracted before calculating the correlation value.

To address the temporal relationship between the activity of PCs and PC-MLIs, we developed the following analysis; this used only sessions with simultaneously recorded eyelid PCs and PC-MLIs. First, since both PC and MLI activity is precisely temporally connected with behavior, we calculated average firing rates, aligning each trial to the time of CR. This procedure reduced the influence of CS-related changes in activity on the results, phasic responses to the CS for example. Depending on ISI, we used either 100 ms (ISI 250) or 150 ms (ISIs 500, 700, and 750) after CR onset time. Trials with CR onsets occurring earlier that these values before the US were excluded from the analysis, as they would have contaminated the results with US-related responses. Second, for both PCs and MLIs we calculated the fraction of the full decrease or increase respectively that happened prior to CR onset. A value of 0 with this analysis indicates that there was no change in that cell's response prior to CR onset, while a value of 1 indicates that the maximum change in firing happened prior to CR onset. Due to this measure being dimensionless it provides the same type of normalization for both cell types, thus, we used it to find whether PCs or MLIs typically change their activity first before CR onset.

For spike-triggered cross-correlogram analysis we aligned spike times of unit 1 within a ±window onto the spike times of unit 2 (1 ms bins, ±30 ms). The process was repeated for all spikes of unit 2 during 10 s before each trial onset, to produce a spike count of unit 1 as a function of time. The significance of modulation in cross-correlograms was assessed by performing the same analyses on shuffled spike times, making unit 1 and unit 2 independent, and computing the mean and standard deviation of cross-correlogram spike count of shuffled data. Normalized spike count, shown in *Figure 8* and *Figure 8—figure supplement 1*, was computed by dividing spike count of unit 1 by the mean of the shuffled data. Z-scores were then computed to identify significantly modulated pairs of PCs and MLIs. Significantly modulated pair was defined as having at least 2 time bins, excluding $t=0$ bin, reaching above 95% significance thresholds ($Z = \pm 3.34$, corrected for multiple comparisons with Bonferroni correction). The procedure of the pause/burst-triggered cross-correlogram analysis was similar to the spike triggered, except data was aligned only to the largest (or the smallest) 35% (50% for simulation) of PC inter-spike intervals, which were operationally defined as pauses and bursts of PCs activity, respectively. Significantly modulated pairs were defined as having by at least 2 significant bins more after $t=0$ compared to before $t=0$ epoch.

## Grouping

To further investigate how MLIs correspond to the onset and amplitude of conditioned eyelid responses, data from each ISI were divided into equal subgroups of trials by CR onset in relation to CS onset or by CR amplitude. Single-unit data were divided with respect to CR onset and amplitude in order to further differentiate unit activity during CRs with different onsets and amplitudes due to the differing amount of variability observed in each response measure. For each ISI, the data for each subgroup was divided equally so the same percentage of trials exists within each CR onset range or amplitude range. A few exceptions to dividing the data within each ISI equally were unavoidable and mostly involved the percentage of data in the group representing the non-CR trials within each ISI. Dividing the data with respect to CR onset involved aligning each trial by CS onset and sorting the behavioral data into equal groups with respect to the CR onset time. Subgroup eyelid CRs and MLI data was then averaged within each training paradigm along with non-CR trials and overlaid to investigate how MLI activity relates to differently timed CRs. Dividing the data with respect to CR amplitude involved aligning each trial by CS onset and sorting the behavioral data into equal subgroups with respect to CR amplitude above pre-CS baseline. Subgroup eyelid CRs and single-unit data were then averaged and overlaid to investigate how MLI activity relates to CRs with different amplitudes. The absence of overlap in 95% confidence intervals between groups of average single-unit activity indicated a significant difference.

## Histology

After the conclusion of experiments, final tetrode placement was determined by making small marking lesions by passing 10 µA of anodal DC current for 10 s through tetrodes which yielded data. Animals were killed with an overdose of sodium pentobarbital and perfused intracardially with 0.9% saline (~1.0 L) followed by 10% formalin (~1.0 L). Heads were post-fixed in formalin for at least 3 days after which tetrodes were removed and the brains were extracted. Brains were then cryo-protected in 30% sucrose in formalin for 3 days, embedded in an albumin gelatin mixture, and the cerebellum was sectioned using a freezing microtome at 40 µm. Tissue was mounted on slides and stained with cresyl

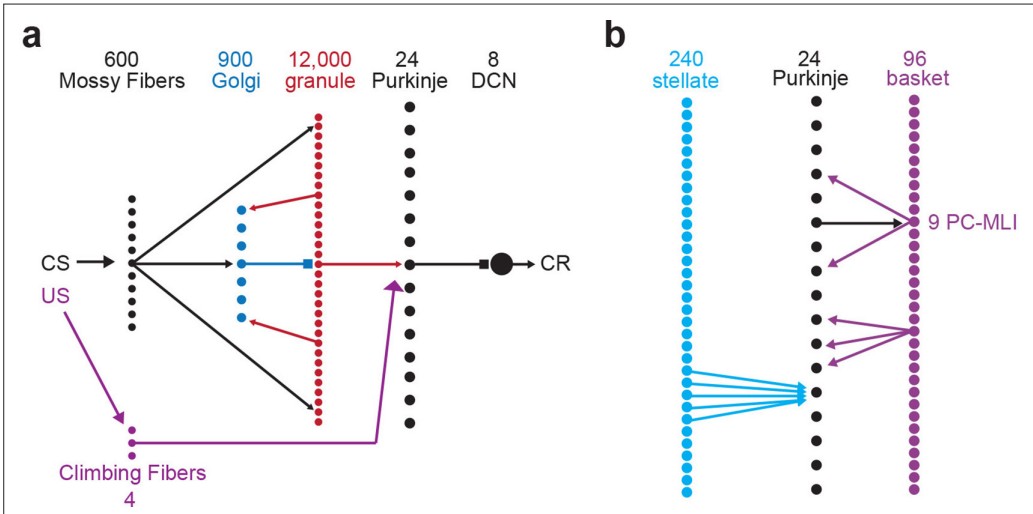

**Figure 14.** Schematic representation of connectivity in the cerebellar simulation. (**a**) Indicates two inputs to the cerebellum -- mossy and climbing fibers -- and a schematic representation of the connectivity of four cerebellar neuron types. The number of each cell type represented in the simulation are indicted by numbers at top. Mossy fiber inputs diverge to excite Golgi and granule cells and converge to excite deep cerebellar neurons (DCN); the latter is not shown on the diagram. Granule cells excite Purkinje and Golgi cells, while Golgi cells inhibit nearby granule cells. (**b**) Schematic representation of the connectivity between molecular layer interneurons (MLIs) and Purkinje cells (PCs). Stellate cells converge to inhibit PCs. PCs inhibit a subset of basket cells near them (the PC-MLIs) and basket cells inhibit nearby PCs, excluding those that could form reciprocal connections.

violet, sections were then examined to determine the final location of each tetrode and this depth was compared with depth records from turning sessions to identify the location of unit recordings (*Figure 13a*).

## Computer simulation of cerebellum circuitry

We employed a computational simulation intended to emulate the synaptic organization and physiology of the cerebellum (*Eccles et al., 1967*; *Hull and Regehr, 2022*; *Figure 5a*). This simulation has been employed over many years and has been optimized to successfully model many aspects of cerebellar processing, including bidirectional learning, adaptive timing, and roles for multiple sites of plasticity, short-term plasticity, and recurrent feedback (*Medina and Mauk, 1999*; *Medina et al., 2000*; *Medina et al., 2002*; *Ohyama et al., 2010*; *Kalmbach et al., 2011*; *Khilkevich et al., 2018*). The details of this simulation have been presented elsewhere (*Medina and Mauk, 1999*; *Medina and Mauk, 2000*; *Medina et al., 2000*; *Medina et al., 2001*; *Medina et al., 2002*; *Ohyama et al., 2010*; *Kalmbach et al., 2011*; *Li et al., 2013*; *Khilkevich et al., 2018*) and source code is available at https://github.com/mauk-lab-utexas/CBMSim, (*Halverson, 2022*; copy archived at swh:1:rev:6f-592845581c0f06bd80a9bd43dabba2000965bf). Here, we will briefly summarize the simulation. As an approximation to the ratio of cell types within the cerebellum, the simulation implemented the following neuronal elements: 600 mossy fibers, 12,000 granule cells, 900 Golgi cells, 24 PCs, 8 deep cerebellar nuclei cells and 4 climbing fibers. (*Figure 14a*). Further, connections between PCs and MLIs were simulated via 96 basket cells (including 9 designated as PC-MLIs) and 240 stellate cells (*Figure 14b*). These neurons were interconnected to emulate a parasagittal stripe, where all PCs receive input from climbing fibers of the same type – that is, those activated by an eyelid US and that control eyelid responses with their output. Each neuron was represented as a conductance-based calculation of membrane potential, with spikes determined by thresholds that varied according to recent spiking activity to implement the net effects of active conductances. Using 1 ms time steps, the change in membrane potential was calculated according to synaptic and leak conductances and to membrane capacitance. Spikes were initiated when membrane potential exceeded threshold. Threshold increased for each spike and returned exponentially to the baseline value. All synaptic conductances were based on time constants derived from in vitro studies; the threshold properties of each representation was tuned to produce behavior that matched the in vivo characteristics of the target cell type.

The simulation represented the geometric relationships and divergence and convergence ratios of synaptic connections within the cerebellum by generating a two-dimensional array of granule cells, Golgi cells, and mossy fiber glomeruli. For each type of connection an eligibility span was specified to represent the region of the array that the presynaptic neuron could potentially make contact with a postsynaptic target. These spans are based on published accounts of the spatial relationships of connections within the cerebellum. For example, since the axons of granule cells run transversely through the cerebellar cortex, the contact area for granule cells was a narrow rectangle. While this area constrained the range over which cells could make connections, the connections were determined randomly in a way that produced the known divergence and convergence ratios (e.g. each granule cell could only receive four mossy fiber inputs).

Separate rules for plasticity were implemented at two types of synapses within the simulation: (1) the granule cell-to-PC synapses and (2) the mossy fiber-to-DCN synapses. A granule cell-to-PC synapse underwent LTD or LTP every time it fired a threshold burst of spikes, LTD occurred when this burst fell within a window between 300 and 100 ms prior to a climbing fiber input to the PC, otherwise LTP occurred. Mossy fiber-to-DCN synapses active within a time window of an abrupt pause in PC activity underwent LTD whereas those active during strong Purkinje activity underwent LTP.

Eyelid conditioning was simulated by presenting to the simulation mossy fiber and climbing fiber inputs based on empirical recordings during eyelid conditioning. Each mossy fiber was assigned a background firing rate between 1 and 40 Hz. To mimic activation of mossy fibers as a CS, a randomly selected 3% of the mossy fibers were designated phasic CS mossy fibers and were active for a brief 100 ms burst at CS onset. Another randomly selected 3% were designated tonic CS mossy fibers and were activate at a rate between 80 and 100 Hz throughout the duration of the CS. All mossy fiber activity was stochastic with the target firing rate for any given time used to determine the probability of activating an excitatory conductance in ways that made the actual activity noisy. The activation of

an excitatory conductance for the four climbing fibers served to mimic the presentation of the US. The averaged and smoothed activity of the eight deep nucleus neurons was used to represent the output of the simulation and the predicted 'eyelid response' of the simulation. These simulations also included excitatory recurrent collaterals from the deep cerebellar nuclei, which deliver to the cerebellar cortex – as mossy fiber input – a copy of output that can be used for sequence learning (*Giovannucci et al., 2017*; *Khilkevich et al., 2018*). These collaterals were included in the same proportion as mossy fibers representing the CS: 5.2% of all mossy fibers (*Houck and Person, 2015*). To calculate the distribution of cross-correlations between PCs-basket cells activity, we repeated 1000 random draws of the same number of PC-BC pairs from the simulation, as we have total (64) in experimental data for ISIs 500, 700, and 750. These randomly drawn pairs were used to calculate cross-correlation values between PCs and basket cells average firing rates in the same fashion as with real data. For statistical comparison between correlation distributions of real data and simulations, we drew 1000 times from the simulation the same amount of PC-PC-MLI pairs as in real data and made a comparison using either paired Student's t-test or two-sample Kolmogorov-Smirnov test for each draw.

## Acknowledgements

We thank K Chung for excellent technical assistance and P Teo and M Yeow for administrative support.

## Additional information

### Funding

| Funder | Grant reference number | Author |
| --- | --- | --- |
| Ministry of Education - Singapore | MOE2016-T2-1-097 | George J Augustine |
| Ministry of Education - Singapore | MOE2017-T3-1-002 | George J Augustine |
| National Institute of Mental Health | MH46904 | Michael D Mauk |
| National Institute of Mental Health | MH74006 | Michael D Mauk |

The funders had no role in study design, data collection and interpretation, or the decision to submit the work for publication.

### Author contributions

Hunter E Halverson, Conceptualization, Data curation, Software, Formal analysis, Validation, Investigation, Visualization, Methodology, Writing – original draft, Writing – review and editing; Jinsook Kim, Conceptualization, Data curation, Formal analysis, Validation, Investigation, Visualization, Methodology, Writing – original draft, Writing – review and editing; Andrei Khilkevich, Data curation, Formal analysis, Validation, Investigation, Visualization, Writing – original draft; Michael D Mauk, Conceptualization, Resources, Software, Supervision, Funding acquisition, Validation, Investigation, Methodology, Writing – original draft, Project administration, Writing – review and editing; George J Augustine, Conceptualization, Resources, Supervision, Funding acquisition, Validation, Investigation, Methodology, Writing – original draft, Project administration, Writing – review and editing

### Author ORCIDs

Jinsook Kim http://orcid.org/0000-0001-6487-6608
George J Augustine http://orcid.org/0000-0001-7408-7485

### Ethics

All mouse procedures were conducted according to the Institutional Animal Care and Use Committee guidelines of the Biopolis Biological Resource Center (AUP 120707) and the Nanyang Technological University (AUP 18095). Treatment of rabbits and surgical procedures were in accordance with National Institutes of Health guidelines and an institutionally approved animal welfare protocol (AUP

2015-00137). All surgery was performed under anesthesia and every effort was made to minimize suffering.

## Decision letter and Author response

Decision letter https://doi.org/10.7554/eLife.77603.sa1
Author response https://doi.org/10.7554/eLife.77603.sa2

## Additional files

### Supplementary files

• Transparent reporting form

### Data availability

All data generated or analysed during this study are included in the manuscript and supporting file; Source data files have been provided. Source code is available at https://github.com/mauk-lab-utexas/CBMSim (copy archived at swh:1:rev:6f592845581c0f06bd80a9bd43dabba2000965bf).

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
