## [Editor Report]

This is an important paper that describes and models an inhibitory pathway that mediates delay conditioning using cerebellar mechanisms in mice and rabbits. The manuscript provides convincing evidence for the proposed mechanisms, further supported by models.

---

## [Decision Letter]

**Decision letter after peer review:**

Thank you for submitting your article "Feedback Inhibition Underlies New Computational Functions of Cerebellar Interneurons" for consideration by *eLife*. Your article has been reviewed by 2 peer reviewers, and the evaluation has been overseen by a Reviewing Editor and Ronald Calabrese as the Senior Editor. The following individual involved in the review of your submission has agreed to reveal their identity: Christian Hansel (Reviewer #3).

Essential revisions:

1. Can the authors make the manuscript more concise, in particular, to better explain the functional role of PC-MLI interactions?

2. It would be helpful to see more comparisons between the experiments and simulation outcomes. Some have been suggested in the detailed comments.

3. The paired recordings are few in number. Can the authors specifically discuss the caveats from this?

4. Can the authors better substantiate the temporal order of PC and PC-MLI firing during CRs?

5. The methods section lacks details, even if parts have been published, there should be a summary. Some points are mentioned by the reviewers. Additionally, can the authors provide a schematic of the simulation?

6. It is journal policy to provide the entire simulation source code and configuration files on an open-source public platform, such as ModelDB, GitHub, or OpenSourceBrain. This should be done at submission time so that the reviewers can see it.

*Reviewer #1 (Recommendations for the authors):*

Specific comments:

Given the variability in IPSC amplitudes, how variable are the maps (Figure 3) from trial to trial? How were presynaptic PC somata located (Figure 3f)? If somata could be located, was the estimate of 1-2 PCs impinging on one MLI tested via somatic stimulation?

Figure 1c, bottom, and Figure 2c: Are the averages shown from different trials in the same cell or in different cells?

Figure 2a,b: Show representative traces for different frequencies of stimulation.

Figure 2 title: Can this be called 'Frequency-independent depression' when the extent of depression is maintained by frequency-dependent mobilization?

Figure 2e: Need a statistical test for a slope of 0.

Page 9: '…but this variability was not due to variations in photostimulation of presynaptic PCs' – This needs to be substantiated with AP probability data from light-stimulated PCs.

Page 11: Non-reciprocal connections- how many PC somata were located within the light spot that evoked IPSCs? The paired recordings are technically very challenging, yet, a sample size of 6 is too small to conclude that reciprocal connections do not exist, especially since PC-MLIs are also a small fraction.

Figure 4e: Was this analysis done on evoked or spontaneous spikes in PCs?

Figure 5b: What are the light grey and black bars?

Figure 5g and S3: The dip in MLI firing is symmetrical at about 0. This is not the case for the simulation.

Figure 5i,h: Are the curves showing averages of all the PC-MLIs recorded? If yes, show the error as well.

Figure 6: Wouldn't these results be expected given that PC-MLIs are identified based on a strong correlation to CR kinematics?

Figure 7: all data in this figure are representative. Need group data with appropriate statistics.

Page 23: "These comparisons indicate that the anti-correlated activity of eyelid PCs and putative PC-MLI arises from inhibitory feedback from PCs to PC-MLI."

This is a strong statement. Data in Supp. Figure 4 are consistent with this interpretation but they do not indicate it.

Page 23: "For trials with CRs, PCs reached 50% of their maximum response before PC-MLIs did" – Why is this indicative of earlier PC response? Responses seem to start at about the same time, with similar slopes, but MLIs show a bigger modulation than PCs. Consequently, 50% max response occurs later in MLIs compared to PCs.

Figure 9: How does the presence of PC to MLI synapses lead to lesser overall plasticity between PFs and PCs?

*Reviewer #2 (Recommendations for the authors):*

1) In the recordings from connected PC-MLI pairs, some parameters are pulled from very low numbers of recordings (n<10). For example, conclusions discussed on p. 5 and illustrated in Figure 1d (and following panels) are based on 7 recordings. It is understandable that it is difficult to find connected pairs, yet as a result, the statistical power is low. This should at least be discussed as a caveat.

2) Figure 5: can the computer simulation predict how many PC-modulated MLIs need to be involved to enable optimal functioning of the feedback circuit (e.g. for one target PC)?

3) Figure 5b3: how is the Gaussian distribution peaking at an x-axis value of 0.5 determined? It seems that there is a separate, distinct peak at about 0.3.

4) Figure 7: This is a very interesting figure that shows that the onset of the PC pause occurs before the peak of MLI activity. If we assume that MLI firing contributes to prolonging the pause, what then mediates the early pause component? Is that PF-PC LTD? Along those lines, is there a necessity/involvement of LTP?

5) On p. 31, last paragraph, the authors state that '…..suggested by our simulations is to require a smaller net change in excitatory synaptic input for PCs to decrease their activity to the level required to produce a well-timed CR during the CS'. This observation resonates well with the recent finding that under realistic [ca^2+^]o and [Mg^2+^]o conditions (1.2mM/1mM instead of the more classically used, but incorrect 2mM/2mM) plasticity conditions are less permissive for LTD (Titley et al., J. Physiol. 597, 2019). If correct, those findings suggest that there are plenty of activity conditions under which LTP is induced, but LTD requires a narrow range of specific temporal activity signatures. In light of this, the findings from the computer simulation are particularly meaningful. The authors might want to add this aspect to their discussion.

[Editors’ note: further revisions were suggested prior to acceptance, as described below.]

Thank you for resubmitting your work entitled "Feedback Inhibition Underlies New Computational Functions of Cerebellar Interneurons" for further consideration by *eLife*. Your revised article has been evaluated by Ronald Calabrese (Senior Editor) and a Reviewing Editor.

The manuscript has been improved but there are some remaining issues that need to be addressed, as outlined below:

1. The reviewers agree that the revision has been done well and most key aspects have been addressed.

2. There are a few small changes that the reviewers suggest to improve clarity and improve the discussion.

3. The provided GitHub link provides a source dump, which is good, but as presented does not make it easy for a reader to replicate or understand the analysis or figures. It needs documentation. Can the authors provide a systematic README which explains how a user may use their code to replicate their results? Ideally this should provide scripts and how they are invoked to replicate the relevant figures.

*Reviewer #2 (Recommendations for the authors):*

The revised version is a delight to read. Though longer than the original submission, the authors have done a fantastic job in interconnecting the various elements (slice physiology, in vivo physiology and computational simulations) and motivating one with the other. This manuscript is a rigorous and exciting dissection of the PC to PC-MLI pathway and its role in motor learning. This manuscript will be an important addition to the cerebellar circuits literature. The authors have addressed most of my comments to my complete satisfaction.

A few issues remain and once addressed, these can be checked at the editorial level for faster acceptance:

1. Line 326 – "…timing of CRs (Figure 5b3)" – but Figure 5b shows correlations with eyelid position and not with CR timing?

2. Figure 8a and corresponding results text – line 379 – the very small suppression window in vivo during conditioning is at odds with the other results (slice, simulation, in vivo intertrial intervals). Is this window in 8a wider than the duration of the merged signal at time 0?

3. Lines 401-406: This part is a little confusing as the authors state that the intertrial intervals were analyzed to avoid large changes in activity during conditioning. Then they also state that large changes in activity during pauses and bursts show the suppression. So why wouldn't that be seen during conditioning as well? Are pauses and bursts seen during the intertrial interval similar to those during the conditioning?

4. I couldn't locate the n for the number of PC-PC-MLI pairs analysed.

5. Figure colors: Perhaps indicate the CS window with a different color since the grey masks the grey of SEM in some cases (Figure 9-Figure suppl. 1 is one example). Also, in Figure 10, it is difficult to discern the dark blue and black. Red and green are not color-blind friendly.

*Reviewer #3 (Recommendations for the authors):*

The authors have only partially addressed my prior concerns and referred to potential future work when at least some discussion of these points would have been appropriate.

---

## [Author Response]

Essential revisions:1. Can the authors make the manuscript more concise, in particular, to better explain the functional role of PC-MLI interactions?

We acknowledge that parts of the previous manuscript were cumbersome. In response, we have heavily edited the sections describing the computational modelling and in vivo recording experiments. We are confident that our efforts have improved the clarity of the text and now do a much better job of emphasizing the key points of our study. Hopefully all readers will now find it easy to comprehend the role of feedback inhibition from PCs to PC-MLIs.

2. It would be helpful to see more comparisons between the experiments and simulation outcomes. Some have been suggested in the detailed comments.

We have revised the Results section to make more comparisons between the model and the in vivo recording data and to clarify the relationship between these two types of analysis. In particular, we list several predictions that emerge from the simulation (p. 11, para. 2) and describe in vivo recording experiments that directly test (and confirm) these predictions (pp. 11-17). This new approach, entirely motivated by the reviewers’ comments, definitely improves the logical flow of the paper and strengthens our conclusions.

3. The paired recordings are few in number. Can the authors specifically discuss the caveats from this?

We must begin by clarifying that we obtained a total of 193 paired recordings; this is by no means a small number! However, despite investing more than 2 years in this experiment, out of these 193 recordings only 10 demonstrated synaptic coupling between PCs and identified PC-MLIs: 7 PC → PC-MLI connections (Figure 4a) + 3 PC-MLI → PC connections (Figure 4b). In none of these 10 cases did we observe reciprocal coupling; presumably this is the point that this comment is addressing.

While the 0/10 cases provide strong evidence for a lack of reciprocal connectivity, we acknowledge that it is still possible there is a low rate of reciprocity that was missed by having only 10 synaptically-coupled pairs. We were unable to find any applicable statistical model to consider this possibility. Thus, we have addressed the issue by simulating the statistical probabilities involved, using a random number generator and a conservative assumption of 100 coupled PC and PC-MLI pairs in a slice. Our findings are plotted in Author response image 1. In brief, a 0.5 rate of reciprocal connectivity between PC and PC-MLI pairs would almost never yield our observation of 10 pairs with no reciprocal coupling. If there is a 0.1 rate of reciprocal connectivity, then there is a 42% chance of observing no reciprocal connections out of 10; if the reciprocal connectivity is reduced to 0.01, there is a 93% chance of obtaining our observed results. A 5% chance of obtaining our results would occur at a rate of reciprocal connectivity of approximately 0.2.

**Author response image 1. sa2fig1:** 

In light of this analysis, we conclude that our current results indicate that there is, at most, a low probability that the inhibitory circuits between PCs and MLIs are reciprocal. It is much more likely that these connections are non-reciprocal.Therefore, we have addressed the reviewers’ concerns by revising our text (p 9, para. 2) as follows:

Overall, in a total of 193 paired recordings, only unidirectional connections (either PC-to-MLI or MLI-to-PC) or unconnected pairs were detected; in no case were reciprocally connected PCs and MLIs observed (Figure 4c). Although we cannot completely exclude the possibility of reciprocal connections, due to the limited number of recordings from connected pairs, our findings of no reciprocally connected cells in 10 connected PC/PC-MLI pairs indicate it is unlikely that these connections are reciprocal. Simulations indicate that our results would almost never be obtained (probability less than 0.001) if 50% of PC/PC-MLI connections were reciprocal, a 0.05 probability of observing no reciprocal connections in 10 trials if 20% of all connections are reciprocal and a probability greater than 0.9 if 1% or less of the connections are reciprocal. In summary, our data indicate that few, if any, PC/PC-MLI connections are reciprocal; thus, PCs will influence other PCs primarily through disinhibition, while not influencing their own behavior through recurrent inhibition (Figure 4d).

4. Can the authors better substantiate the temporal order of PC and PC-MLI firing during CRs?

In our revised manuscript, we have addressed this issue by creating a new section that is entirely devoted to analysis of the temporal relationship between PC and PC-MLI activity (pp. 14-17). In brief, here are the main lines of evidence that PCs fire prior to PC-MLIs, both in baseline conditions and during conditioned eyelid responses:

During baseline activity, we have provided 3 types of evidence that PCs fire prior to putative PC-MLIs:

1) A spike-triggered average of PC and putative PC-MLI activity during baseline firing showed a modest decrease in PC-MLI firing rate in response to a PC action potential (Figure 8a; see also Figure 8—figure supplement 1a and 1b).

2) A pause in PC activity caused a very substantial rise in activity in putative PC-MLIs (Figure 8c; see also Figure 8—figure supplement 1c).

3) A burst of PC activity caused a decline in putative PC-MLI activity (Figure 8d; see also Figure 8—figure supplement 1d).

We have an additional 3 lines of evidence showing that PCs fire prior to putative PC-MLIs during CRs:

1) Simultaneous recordings of the time course in changes in PC and putative PC-MLI activity during CRs indicate that PC activity usually declined prior to the activity of putative PC-MLIs. This is clearly visible in the examples shown in Figure 9c, and is discernible in the averaged data shown in Figure 9—figure supplement 1a.

2) We measured the delay between the time at which PC activity reached 50% of its minimum during the CS and the time at which the activity of putative PC-MLIs reached 50% of its maximum during single trials. Strikingly, whenever CRs were observed, PCs reached their half-maximal response before putative PC-MLIs did (Figure 9─figure supplement 1b).

3) We also measured the collective timing of changes in the activity of putative PC-MLIs and eyelid PCs during conditioning across all of our paired recordings. This was done by calculating a ratio representing the magnitude of changes in activity prior to CR onset, normalized to the peak amplitude of the change during the entire interval. The distribution of differences in the timing of changes in PC and PC-MLI activity has a mean that is greater than zero (Figure 10), indicating that eyelid PCs decreased their activity before putative PC-MLIs increased their activity in a majority of cases.

To make these points more clearly, we have significantly revamped the in vivo recording section of the Results.

5. The methods section lacks details, even if parts have been published, there should be a summary. Some points are mentioned by the reviewers. Additionally, can the authors provide a schematic of the simulation?

In response to this comment, we have revised the Methods section to include much more detail about the simulation methods, including two schematic diagrams. The methods used for the non-computational work in the paper are already described in extensive detail.

6. It is journal policy to provide the entire simulation source code and configuration files on an open-source public platform, such as ModelDB, GitHub, or OpenSourceBrain. This should be done at submission time so that the reviewers can see it.

Source code for the simulation is now available on GitHub: https://github.com/mauk-lab-utexas/CBMSim.

Reviewer #1 (Recommendations for the authors):Specific comments:Given the variability in IPSC amplitudes, how variable are the maps (Figure 3) from trial to trial?

Although the amplitude of IPSCs evoked at a given pixel were quite variable, the spatial range of synaptic input maps varied minimally from trial to trial. That is because the pixel-to-pixel variability derives from the inherent variability of transmission (but few failures) at the PC-to-PC-MLI synapse, rather than fluctuations in light-evoked firing of PCs. To obtain the most reliable estimate of IPSC amplitude, the maps shown in Figures 3b-d are averages of 3 trials.

How were presynaptic PC somata located (Figure 3f)?

The “estimated” location of PC somata in Figure 3f was based on the assumption that PC somata were located at the center of the input field, when projected down to the Purkinje cell layer. This was only an estimate, but is likely to be accurate to within 20-40 µm (i.e. less than the width of the histobars in Figure 4f).

If somata could be located, was the estimate of 1-2 PCs impinging on one MLI tested via somatic stimulation?

Recall that we successfully patched 7 presynaptic PC-postsynaptic PC-MLIs pairs in 2 years of recordings (pp. 8-9). This high degree of difficulty precluded determining whether there were additional presynaptic PCs, beyond the one that we were very fortunate to record from. More generally, the whole point of our optogenetic circuit-mapping technology is to avoid having to ferret out the location of presynaptic neurons via patching!

Figure 1c, bottom, and Figure 2c: Are the averages shown from different trials in the same cell or in different cells?

They are from different cells.

Figure 2a,b: Show representative traces for different frequencies of stimulation.

We have added these traces in Figure2—figure supplement 1.

Figure 2 title: Can this be called 'Frequency-independent depression' when the extent of depression is maintained by frequency-dependent mobilization?

It is called “frequency-independent” because synaptic strength was maintained at approximately half of its initial strength – even during prolonged activity – independent of stimulus frequency. This differs from virtually every other type of synapse in the brain, where the degree (and rate) of depression strongly depends upon the frequency of activity. While apt, “frequency-independent” is not our original terminology: it was coined by Turecek et al. (2016), to describe identical behavior at the PC-to-DCN synapse.

Our contribution is to show that frequency-dependent vesicle mobilization enables the frequency-independence of depression. One way to view it is that two frequency-dependent processes – vesicle depletion (negative) and mobilization (positive) – cancel each other out to create frequency-independent synaptic transmission. This balance is remarkable and is remarkably important for PCs, which tonically fire at high frequencies.

Figure 2e: Need a statistical test for a slope of 0.

The slope of regression was not significantly different from 0 (p = 0.85, F-test). This information has been added to the figure legend.

Page 9: '…but this variability was not due to variations in photostimulation of presynaptic PCs' – This needs to be substantiated with AP probability data from light-stimulated PCs.

It is hardly controversial for us to say that ChR2 reliably photostimulates PCs. Indeed, we state this on the last line of p. 3: “photostimulation reliably evoked action potentials in every PC examined (20/20)”. Further, this is documented in the time domain in Figure 1b and in the spatial domain in Figure 3a. Thus, we respectfully suggest that it is unreasonable for the reviewer to ask that we show additional – and fairly pedantic – data, particularly after complaining that there is already too much data in the paper!

Page 11: Non-reciprocal connections- how many PC somata were located within the light spot that evoked IPSCs?

We have addressed this point at the bottom of page 8:

“although our optogenetic mapping identified the volume in which presynaptic PCs were located, there was still a low likelihood of sampling the 1 or 2 presynaptic PCs within the dozens of PCs within this volume”.

The paired recordings are technically very challenging, yet, a sample size of 6 is too small to conclude that reciprocal connections do not exist, especially since PC-MLIs are also a small fraction.

Out of 193 paired recordings of PCs and MLIs, only 10 demonstrated synaptic coupling between PCs and identified PC-MLIs: 7 PC → PC-MLI connections (Figure 4a) + 3 PC-MLI → PC connections (Figure 4b). (Note that the number is 10, rather than 6, because we did additional data analysis.) In none of these 10 cases did we observe reciprocal coupling.

While the 0/10 cases provide strong evidence for a lack of reciprocal connectivity, we acknowledge that it is still possible there is a low rate of reciprocity that was missed by having only 10 synaptically-coupled pairs. We were unable to find any applicable statistical model to consider this possibility. Thus, we have addressed the issue by simulating the statistical probabilities involved, using a random number generator and a conservative assumption of 100 coupled PC and PC-MLI pairs in a slice. Our findings are plotted in Author response image 1. In brief, a 0.5 rate of reciprocal connectivity between PC and PC-MLI pairs would almost never yield our observation of 10 pairs with no reciprocal coupling. If there is a 0.1 rate of reciprocal connectivity, then there is a 42% chance of observing no reciprocal connections out of 10; if the reciprocal connectivity is reduced to 0.01, there is a 93% chance of obtaining our observed results. A 5% chance of obtaining our results would occur at a rate of reciprocal connectivity of approximately 0.2.

In light of this analysis, we conclude that our current results indicate that there is, at most, a low probability that the inhibitory circuits between PCs and MLIs are reciprocal. It is much more likely that these connections are non-reciprocal.

Therefore, we have addressed the reviewer’s concern by revising our text (p 9, para. 2) as follows:

Overall, in a total of 193 paired recordings, only unidirectional connections (either PC-to-MLI or MLI-to-PC) or unconnected pairs were detected; in no case were reciprocally connected PCs and MLIs observed (Figure 4c). Although we cannot completely exclude the possibility of reciprocal connections, due to the limited number of recordings from connected pairs, our findings of no reciprocally connected cells in 10 connected PC/PC-MLI pairs indicate it is unlikely that these connections are reciprocal. Simulations indicate that our results would almost never be obtained (probability less than 0.001) if 50% of PC/PC-MLI connections were reciprocal, a 0.05 probability of observing no reciprocal connections in 10 trials if 20% of all connections are reciprocal and a probability greater than 0.9 if 1% or less of the connections are reciprocal. In summary, our data indicate that few, if any, PC/PC-MLI connections are reciprocal; thus, PCs will influence other PCs primarily through disinhibition, while not influencing their own behavior through recurrent inhibition (Figure 4d).

Figure 4e: Was this analysis done on evoked or spontaneous spikes in PCs?

Our analysis was done on both evoked and spontaneous spikes in PCs. This information has been added to the figure legend.

Figure 5b: What are the light grey and black bars?

Figure 5b1-2 shows simulation results both with (2) and without (1) PC feedback to PC-MLIs. For panel 1 the gray (now white) bars indicate the stellate cells and the black (now gray) bars indicate basket cells (no PC-MLIs in this simulation). For panel 2 the gray (now white) bars represent both stellate and basket cells and the black (now gray) bars represent PC-MLIs.

Figure 5g and S3: The dip in MLI firing is symmetrical at about 0. This is not the case for the simulation.

We acknowledge that Figures 8a and 8b (new numbering system) are not identical. Probably the build-up of activity in the simulation during the few milliseconds prior to 0 represents PF EPSPs, which evoked action potentials in both PCs and MLIs. As discussed in the text, there is much less background activity in the simulation compared to in vivo conditions. This might cause the build-up to be more evident in the simulation. The main point is that no model is perfect and the simulation is intended to reproduce only qualitative aspects of the in vivo data, in this case showing that a single PC simple spike can impact MLI firing both in vivo and in simulation. We acknowledge the differences between the in vivo data and simulation data in the text (p. 15, para. 1).

Figure 5i,h: Are the curves showing averages of all the PC-MLIs recorded? If yes, show the error as well.

These figures (now Figures 8c and d) show individual examples recorded on the same tetrode.

Figure 6: Wouldn't these results be expected given that PC-MLIs are identified based on a strong correlation to CR kinematics?

It is true that we used the correlation with eyelid position to operationally define PC-MLIs; specifically, we used session averages between eyelid position and firing rate. However, the analysis shown in the figure (now Figure 7) indicates that the tight relationship with PC-MLI activity includes not only eyelid position, but also eyelid velocity; it also applies at 3 different ISIs. Further, this relationship extends to other kinematic variables: sorting the data by CR onset and amplitude (Figure 7—figure supplement 1) shows that the relationship between PC-MLIs and eyelid CRs is consistent not only across ISIs, but also within each ISI we tested. Thus, PC-MLI activity is closely linked to conditioned eyelid movements. These results are also consistent with previous results grouping eyelid PC activity by CR onset and amplitude (Halverson et al., 2016; Halverson et al., 2018).

Figure 7: all data in this figure are representative. Need group data with appropriate statistics.

Group data are now shown in Figure9—figure supplement 1a, including 95% confidence intervals.

Page 23: "These comparisons indicate that the anti-correlated activity of eyelid PCs and putative PC-MLI arises from inhibitory feedback from PCs to PC-MLI."This is a strong statement. Data in Supp. Figure 4 are consistent with this interpretation but they do not indicate it.

This part of the text, including previous Figure 8a and Supplementary Figure 4, have been removed.

Page 23: "For trials with CRs, PCs reached 50% of their maximum response before PC-MLIs did" – Why is this indicative of earlier PC response? Responses seem to start at about the same time, with similar slopes, but MLIs show a bigger modulation than PCs. Consequently, 50% max response occurs later in MLIs compared to PCs.

As mentioned above, CR-related changes in PC and PC-MLI activity are kinetically complex. 50% of the maximum response is a fairly standard way to quantify the timing of response onset. If the reviewer is using “bigger” to mean a larger response, the 50% criterion is explicitly intended to take into account differences in response amplitude. If the reviewer intends “bigger” to mean peaking at a later time, well that is consistent with our conclusion that PC responses begin earlier than PC-MLI responses. More importantly, 50% of maximum is not the only way we have compared the timing of PC and PC-MLI responses: we also have shown the latency of change in CR-aligned averages, trial-by-trial raster plots or 50% maximum response latency between simultaneous recordings of eyelid PC and PC-MLI. In nearly all cases, eyelid PC activity decreases before both CR onset and the increase in PC-MLI activity.

Figure 9: How does the presence of PC to MLI synapses lead to lesser overall plasticity between PFs and PCs?

This is an interesting question that we cannot answer yet. Clearly, future studies will be necessary to determine the specific role of the PC to PC-MLI connection in regulating cerebellar plasticity and learning. We have added a sentence to the Discussion (p. 24, para. 3) to make this point.

Reviewer #2 (Recommendations for the authors):1) In the recordings from connected PC-MLI pairs, some parameters are pulled from very low numbers of recordings (n<10). For example, conclusions discussed on p. 5 and illustrated in Figure 1d (and following panels) are based on 7 recordings. It is understandable that it is difficult to find connected pairs, yet as a result, the statistical power is low. This should at least be discussed as a caveat.

Out of 193 paired recordings of PCs and MLIs, only 10 demonstrated synaptic coupling between PCs and identified PC-MLIs: 7 PC → PC-MLI connections (Figure 4a) + 3 PC-MLI → PC connections (Figure 4b). (Note that the number is 10, rather than 7, because we did additional data analysis.) In none of these 10 cases did we observe reciprocal coupling.

While the 0/10 cases provide strong evidence for a lack of reciprocal connectivity, we acknowledge that it is still possible there is a low rate of reciprocity that was missed by having only 10 synaptically-coupled pairs. We were unable to find any applicable statistical model to consider this possibility. Thus, we have addressed the issue by simulating the statistical probabilities involved, using a random number generator and a conservative assumption of 100 coupled PC and PC-MLI pairs in a slice. Our findings are plotted in Author response image 1. In brief, a 0.5 rate of reciprocal connectivity between PC and PC-MLI pairs would almost never yield our observation of 10 pairs with no reciprocal coupling. If there is a 0.1 rate of reciprocal connectivity, then there is a 42% chance of observing no reciprocal connections out of 10; if the reciprocal connectivity is reduced to 0.01, there is a 93% chance of obtaining our observed results. A 5% chance of obtaining our results would occur at a rate of reciprocal connectivity of approximately 0.2.

In light of this analysis, we conclude that our current results indicate that there is, at most, a low probability that the inhibitory circuits between PCs and MLIs are reciprocal. It is much more likely that these connections are non-reciprocal.

Therefore, we have addressed the reviewer’s concern by revising our text (p 9, para. 2) as follows:

Overall, in a total of 193 paired recordings, only unidirectional connections (either PC-to-MLI or MLI-to-PC) or unconnected pairs were detected; in no case were reciprocally connected PCs and MLIs observed (Figure 4c). Although we cannot completely exclude the possibility of reciprocal connections, due to the limited number of recordings from connected pairs, our findings of no reciprocally connected cells in 10 connected PC/PC-MLI pairs indicate it is unlikely that these connections are reciprocal. Simulations indicate that our results would almost never be obtained (probability less than 0.001) if 50% of PC/PC-MLI connections were reciprocal, a 0.05 probability of observing no reciprocal connections in 10 trials if 20% of all connections are reciprocal and a probability greater than 0.9 if 1% or less of the connections are reciprocal. In summary, our data indicate that few, if any, PC/PC-MLI connections are reciprocal; thus, PCs will influence other PCs primarily through disinhibition, while not influencing their own behavior through recurrent inhibition (Figure 4d).

2) Figure 5: can the computer simulation predict how many PC-modulated MLIs need to be involved to enable optimal functioning of the feedback circuit (e.g. for one target PC)?

The simulation is capable of answering this and many other related questions about the inhibitory negative-feedback loop. This point is beyond the scope of the current manuscript (which is already 58 pages long), but is fertile ground for future simulation- based experiments. To encourage such efforts, we have made the simulation source code available to readers on GitHub: https://github.com/mauk-lab-utexas/CBMSim.

3) Figure 5b3: how is the Gaussian distribution peaking at an x-axis value of 0.5 determined? It seems that there is a separate, distinct peak at about 0.3.

The distribution of MLI recording correlations with eyelid position was a single skewed distribution that we fit with two Gaussian distributions. The right-skewed component was fit to the data that peaked at 0.43, which was the cut-off point for our operational definition of eyelid PC-MLIs. The determination of the fit for the rest (majority) of the data was established after subtracting off the right-skewed component. Our fits suggest that the small peak around 0.3 is where the two components overlap maximally, though it is formally possible that this represents a small, third component.

4) Figure 7: This is a very interesting figure that shows that the onset of the PC pause occurs before the peak of MLI activity. If we assume that MLI firing contributes to prolonging the pause, what then mediates the early pause component? Is that PF-PC LTD? Along those lines, is there a necessity/involvement of LTP?

We share the reviewer’s enthusiasm for this figure (now called Figure 9). The simulation includes both PF-PC LTP and LTD. The exact timing of the early portion of the pause is probably the balance between preceding LTP and LTD at synapses that become active just before that 400 ms window. The synapses that are activated earliest (furthest in time from US onset) likely undergo LTP, because the majority of their activity falls outside of the plasticity window and delays the PC pause. This LTP essentially delays CR onset by inhibiting deep nucleus activity shortly after CS onset (up to the plasticity window). In contrast, the synapses that are activated very close to the plasticity window will likely undergo LTD and influence the PC pause to happen earlier. We believe this LTD/LTP balance controls CR onset.

5) On p. 31, last paragraph, the authors state that '…..suggested by our simulations is to require a smaller net change in excitatory synaptic input for PCs to decrease their activity to the level required to produce a well-timed CR during the CS'. This observation resonates well with the recent finding that under realistic [ca^2+^]o and [Mg^2+^]o conditions (1.2mM/1mM instead of the more classically used, but incorrect 2mM/2mM) plasticity conditions are less permissive for LTD (Titley et al., J. Physiol. 597, 2019). If correct, those findings suggest that there are plenty of activity conditions under which LTP is induced, but LTD requires a narrow range of specific temporal activity signatures. In light of this, the findings from the computer simulation are particularly meaningful. The authors might want to add this aspect to their discussion.

This is a very interesting point and one that deserves a fuller treatment in a paper investigating (or reviewing) the necessary or optimal conditions for LTD vs. LTP. Since we have no data that directly address this point, we have not included it in the revised manuscript.

[Editors’ note: further revisions were suggested prior to acceptance, as described below.]

Reviewer #2 (Recommendations for the authors):The revised version is a delight to read. Though longer than the original submission, the authors have done a fantastic job in interconnecting the various elements (slice physiology, in vivo physiology and computational simulations) and motivating one with the other. This manuscript is a rigorous and exciting dissection of the PC to PC-MLI pathway and its role in motor learning. This manuscript will be an important addition to the cerebellar circuits literature. The authors have addressed most of my comments to my complete satisfaction.A few issues remain and once addressed, these can be checked at the editorial level for faster acceptance:1. Line 326 – "…timing of CRs (Figure 5b3)" – but Figure 5b shows correlations with eyelid position and not with CR timing?

We have revised this sentence (page 12, para. 2) as follows: As in the simulation, the majority of MLIs had little or no correlation with the kinematics of CRs (Figure 5b3).

2. Figure 8a and corresponding results text – line 379 – the very small suppression window in vivo during conditioning is at odds with the other results (slice, simulation, in vivo intertrial intervals).

We agree that the time course of suppression of PC-MLI activity is surprisingly brief and have explained the difference in the time scale of the slice, simulation and in vivo results in the paragraph that spans pp. 14-15. The following paragraph (p. 15, para. 2) also contrasts the small suppression of individual spikes with the larger suppression of activity related to pauses/bursts of PC activity.

Is this window in 8a wider than the duration of the merged signal at time 0?

Yes, the brief decrease in PC-MLI activity after time 0 in Figure 8a is beyond the merged signal time window. The complete lack of PC-MLI spikes at time 0 shows how the merged signal impacts the analysis. Also, the activity of each pair of PC and MLI recorded on the same tetrode would show a similar decrease in spike count after time 0 if this decrease in activity was the result of bleed-over from the merged signal. This relatively small decrease in activity was specific to the subset of PC/PC-MLI pairs we identified on the same tetrode. We have added a sentence to para. 2 on p. 14 to state this:

This relatively small and brief decrease in activity was specific to PC/PC-MLI pairs identified on the same tetrode.

3. Lines 401-406: This part is a little confusing as the authors state that the intertrial intervals were analyzed to avoid large changes in activity during conditioning. Then they also state that large changes in activity during pauses and bursts show the suppression. So why wouldn't that be seen during conditioning as well? Are pauses and bursts seen during the intertrial interval similar to those during the conditioning?

The pauses/bursts of PC activity during the inter-trial interval are fundamentally different from the pauses during conditioning trials. The pauses during conditioning trials are the result of temporally specific LTD that – together with other eyelid PCs – control the kinematics of the CR. The pauses/bursts during the inter-trial interval are not related to behaviour in the same way. The inter-trial activity has a range of PC pause/burst durations that occur independent of the pauses elicited during conditioning trials. The inter-trial interval data provided an opportunity to investigate the dynamics between PCs and PC-MLIs without the expression of the learned CR. If we only found an inverse relationship between PCs and PC-MLIs during the expression of CRs and not during the inter-trial interval it would suggest that direct connectivity (PC feedback inhibition) might not be mediating this dynamic. However, we found the inverse relationship between neurons during expression of CRs and during the inter-trial interval, which supports our hypothesis that this relationship is the result of PC feedback to MLIs. These results also cast doubt on the idea that PC-MLI correlations with CRs are due to PC-MLI learning-related plasticity since we observed the inverse relationship when CRs were not present.

To clarify things, we have revised para. 2 on p. 17 as follows:

In summary, the activity of PCs and putative PC-MLIs is anti-correlated. This is true both during CRs, as well as at other times, and is particularly evident during large changes in PC activity. Further, several different types of analysis all point to the conclusion that changes in PC activity precede both CR onset and changes in putative PC-MLI activity. All these results are consistent with prediction (3) of the simulation. Collectively, our measurements of the activity of PCs and putative PC-MLIs – and their relationship to CRs – are in line with all 3 predictions of the simulation. Given that the computational predictions require the presence of PC-to-PC-MLI feedback inhibition, we conclude that this feedback circuit likely operates in the cerebellum during motor learning.

4. I couldn't locate the n for the number of PC-PC-MLI pairs analysed.

We have added this information to the legend of Figure 10: Distribution showing relationship between timing of decreases in eyelid PCs activity and increases in PC-MLIs activity at each ISI (black = ISI 250, 76 PC/PC-MLI pairs; blue = ISI 500, 50 pairs; red = ISI 700 and 750, 10 pairs).

We did the same thing for the legend of Figure 7, as well as a brief mention in the text (p. 16, para. 3).

5. Figure colors: Perhaps indicate the CS window with a different color since the grey masks the grey of SEM in some cases (Figure 9-Figure suppl. 1 is one example).

We have revised this figure by using a darker shade of grey to indicate the SEM. This makes it easier to differentiate the SEM from the CS.

Also, in Figure 10, it is difficult to discern the dark blue and black. Red and green are not color-blind friendly.

We have revised this figure by changing the black bars to grey; this makes it easier to differentiate the ISI 200 and ISI 500 data.

Reviewer #3 (Recommendations for the authors):The authors have only partially addressed my prior concerns and referred to potential future work when at least some discussion of these points would have been appropriate.

We have made a good-faith effort to incorporate Christian’s comments and have done most of what he suggested, including adding new data and simulations. Out of his 5 suggestions, the only things we did not include were:

(1) saying something about the calcium/activity requirements for LTD, because we simply could not figure out a way to include this; and

(2) predicting how many PC-modulated MLIs are required for optimal functioning of the feedback circuit, because this is a fairly major task that is better addressed in a separate, detailed study.